# Machine learning-based meta-analysis reveals gut microbiome alterations associated with Parkinson's disease

Stefano Romano [1,2] ✉, Jakob Wirbel [2], Rebecca Ansorge [1,3], Christian Schudoma [2], Quinten Raymond Ducarmon[2], Arjan Narbad [1] & Georg Zeller [2,4,5] ✉

There is strong interest in using the gut microbiome for Parkinson's disease (PD) diagnosis and treatment. However, a consensus on PD-associated microbiome features and a multi-study assessment of their diagnostic value is lacking. Here, we present a machine learning meta-analysis of PD microbiome studies of unprecedented scale (4489 samples). Within most studies, microbiome-based machine learning models accurately classify PD patients (average AUC 71.9%). However, these models are study-specific and do not generalise well across other studies (average AUC 61%). Training models on multiple datasets improves their generalizability (average LOSO AUC 68%) and disease specificity as assessed against microbiomes from other neurodegenerative diseases. Moreover, meta-analysis of shotgun metagenomes delineates PD-associated microbial pathways potentially contributing to gut health deterioration and favouring the translocation of pathogenic molecules along the gut-brain axis. Strikingly, microbial pathways for solvent and pesticide biotransformation are enriched in PD. These results align with epidemiological evidence that exposure to these molecules increases PD risk and raise the question of whether gut microbes modulate their toxicity. Here, we offer the most comprehensive overview to date about the PD gut microbiome and provide future reference for its diagnostic and functional potential.

Parkinson's disease (PD) is the second most common age-related neurodegenerative disease after Alzheimer's disease. Recent estimates suggest a doubling of PD patients every ~ 30 years, which might result in around 12 million patients worldwide by 2050[1]. Only a minority of PD cases are thought to be of purely genetic origin and environmental factors are of crucial importance in disease development[2–4]. A hallmark of PD is the accumulation of Lewy's bodies containing misfolded α-synuclein (αSyn) proteins in the central nervous system (CNS), causing neuron toxicity and death[5]. Specifically, the loss of dopaminergic neurons and consequent decrease in dopamine levels are the molecular mechanisms underlying motor impairments observed in PD patients[5]. However, PD manifests with a plethora of both motor and non-motor symptoms, many of which involve the gastrointestinal (GI) tract[6–8]. Among the latter, gastroparesis, gut inflammation, increased intestinal permeability, and constipation are frequently observed[8] and some of these GI symptoms have been shown to be predictive of PD[7]. Strikingly, GI tract involvement can precede motor symptoms by many years. For example, constipation is among the earliest non-motor

[1]Quadram Institute Bioscience, Norwich Research Park, Norwich, UK. [2]Structural and Computational Biology Unit, European Molecular Biology Laboratory, Heidelberg, Germany. [3]Earlham Institute, Norwich Research Park, Norwich, UK. [4]Leiden University Center for Infectious Diseases (LUCID), Leiden University Medical Center, Leiden, Netherlands. [5]Center for Microbiome Analyses and Therapeutics (CMAT), Leiden University Medical Center, Leiden, Netherlands. ✉e-mail: stfno.rmno@gmail.com; georg.zeller@gmail.com

symptoms and can appear up to twenty years before diagnosis[9]. Moreover, recent evidence has linked GI inflammatory diseases, such as IBD, to PD pathophysiology[10,11]. This relationship between GI health and PD has motivated numerous investigations of the putative roles of the gut microbiome in the disease.

We recently conducted a meta-analysis of gut microbiome studies in PD (based on 16S ribosomal RNA gene amplicon sequencing) and showed that when compared to controls, the gut microbiome of PD patients has some common alterations across patient populations from diverse countries and continents[12]. Although high variability between studies was observed, as often in microbiome meta-analyses[13], the gut microbiome in PD patients is typically depleted in short-chain fatty acid (SCFA) producing bacteria. SCFAs are the end product of the bacterial fermentation of complex carbohydrates and they play a pivotal role in maintaining epithelial barrier integrity and colonic immune homoeostasis. Similar results have been confirmed by independent meta-analyses and more recent shotgun metagenomic studies[14–16]. Nevertheless, there is still limited consensus on the bacterial species and metabolic pathways associated with the disease[15–19]. Identifying microbial taxa and especially metabolic functions associated with PD across sampling populations is essential in order to develop mechanistic hypotheses on how the microbiome could possibly contribute to the disease. This will open doors for designing experiments to mechanistically elucidate a putative impact of gut microbes on PD and for developing strategies to use the microbiome for disease diagnosis, prognosis, and treatment.

To date, PD is diagnosed through clinical assessment of motor symptoms which can appear late in the disease course. Hence, there is a clear need for alternative markers to facilitate early diagnosis. To address this, several attempts have been made to use gut microbiome features for building machine learning (ML) classification models that discriminate PD patients from controls[17,18,20–22], reporting up to 90% classification accuracies. However, we currently do not know whether these high prediction accuracies are observed across datasets from different countries. Specifically, model portability, indicating how well models perform when applied to an independent dataset obtained from another sampling population, has never been investigated in the context of PD. This is, however, relevant as it could reveal features (i.e., bacterial taxa/functions) that consistently discriminate PD from controls, thus informing on the potential generalisation and global applicability of such models. Finally, the combination of multiple datasets in a large-scale meta-analysis could ideally lead to more accurate and robust models for PD classification, and it has so far not been thoroughly explored.

Here, to fill this knowledge gap, we perform a large-scale meta-analysis of the gut microbiome in PD to assess how accurately ML models based on the currently available gut microbiome data can discriminate PD patients from controls. We use both public 16S amplicon sequencing and shotgun metagenomics data to extensively evaluate various ML approaches based on single and combined datasets. We complement this ML analysis by conducting the largest meta-analysis so far on the gut microbiome in PD to establish an updated list of prokaryotic taxa and microbial metabolic functions robustly associated with the disease.

## Results

### Datasets overview and beta diversity analysis

We processed a total of 4489 samples obtained from 22 case-control studies across 11 countries and 4 continents that profiled the faecal microbiome of PD patients and controls using 16S amplicon (16S; 3165 samples) and shotgun metagenomics sequencing (SMG; 1324 samples; Table 1). Altogether, the number of samples we used is up to four times larger than those used in previous PD meta-analyses[12,14,16,22]. This first allowed us to investigate the overall

### Table 1 | Overview of the studies re-analysed in this work

| Study | Accession Id | Data type | #Samples | Country |
|---|---|---|---|---|
| Wallen 2021[29] (2 datasets) | PRJNA601994[£] | 16S | 130/184 CTR; 196/323 PD | USA |
| Zhang 2020[24] | CRA001938[$] | 16S | 137 CTR; 63 PD | China |
| Tan 2021[25] | PRJNA494620[£] | 16S | 96 CTR; 104 PD | Malaysia |
| Heintz-Buschart 2017[110] | PRJNA381395[£] | 16S | 38 CTR; 26 PD | Germany |
| Petrov 2017[92] | Personal communication | 16S | 67 CTR; 95 PD | Russia |
| Qian 2018[228] | PRJNA391524[£] | 16S | 45 CTR; 45 PD | China |
| Keshavarzian 2015[94] | PRJNA268515[£] | 16S | 31 CTR; 34 PD | USA |
| Pietrucci 2019[111] | PRJNA510730[£] | 16S | 72 CTR; 80 PD | Italy |
| Aho 2019[112] | PRJEB27564[£] | 16S | 64 CTR; 67 PD | Finland |
| Weis 2019[95] | PRJEB30615[£] | 16S | 25 CTR; 39 PD | Germany |
| Hopfner 2017[59] | PRJEB14928[£] | 16S | 26 CTR; 29 PD | Germany |
| Nishiwaki 2020[14] | DRA009229[*] | 16S | 137 CTR; 223 PD | Japan |
| Cirstea 2020[96] | Personal communication | 16S | 103 CTR; 197 PD | Canada |
| Lubomski 2022[21] | PRJNA808166[£] | 16S | 81 CTR; 103 PD | Australia |
| Kenna 2021[107] | https://doi.org/10.6084/m9.figshare.14345513[&] | 16S | 47 CTR; 86 PD | Canada |
| Jo 2022[19] | PRJNA742875[£] | 16S | 84 CTR; 88 PD | South Korea |
| Total samples | | 16S | 1367 CTR; 1798 PD | – |
| Bedarf 2017[18] | PRJEB17784[£] | SMG | 28 CTR; 31 PD | Germany |
| Qian 2020[17] | PRJNA433459[£] | SMG | 40 CTR; 40 PD | China |
| Mao 2021[46] | PRJNA588035[£] | SMG | 39 CTR; 39 PD | China |
| Jo 2022[19] | PRJNA743718[£] | SMG | 74 CTR; 82 PD | China |
| Wallen 2022[15] | PRJNA834801[£] | SMG | 234 CTR; 490 PD | USA |
| Boktor 2023[16] (2 datasets) | ERP138197/ERP138199[£] | SMG | 72/67 CTR; 42/46 PD | USA |
| Total samples | | SMG | 554 HC; 770 PD | – |

Sample numbers refer to those used in our study after data filtration. CTR = controls; PD = Parkinson's disease patients. Accession numbers refer to different sources: ENA = £; GSA = $; Figshare = &; DDBJ = *. 16S = 16S ribosomal RNA gene amplicon data; SMG = shotgun metagenomic data.

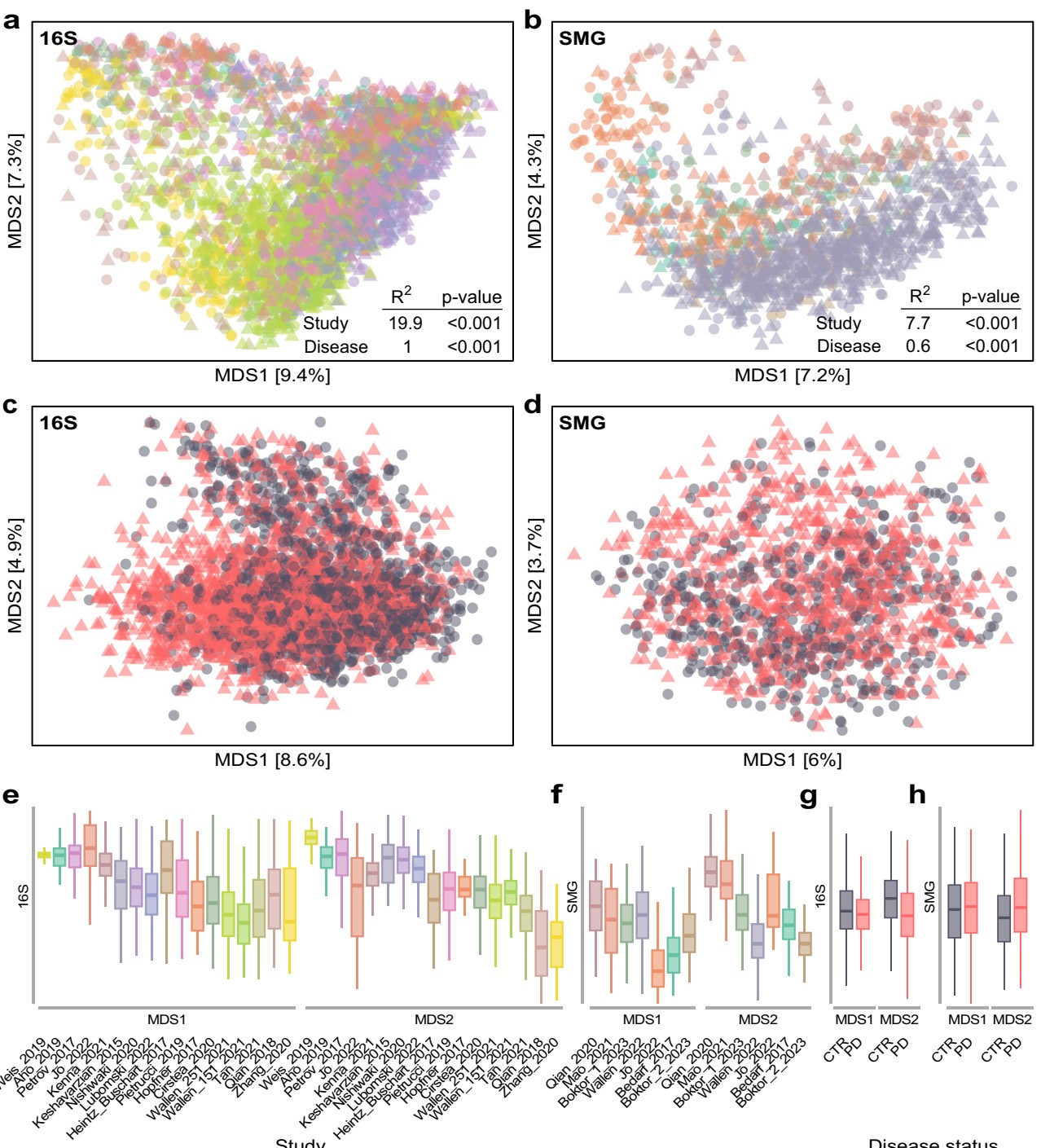

**Fig. 1 | The composition of the gut microbiome significantly differs between PD and controls (CTR).** Distance-based redundancy analysis (dbRDA) was performed on Bray-Curtis dissimilarities calculated for the 16S (**a**, **c**) and SMG data (**b**, **d**) with the percentage of variance explained annotated along the axes. Raw distances (**a**, **b**) and distances conditioned by the study of origin to remove study-specific (batch) effects are shown (**c**, **d**). Boxplots depict the sample distribution along the two components and are coloured to indicate the different datasets used (**e**, **f**) or the disease status (**g**, **h**). Boxes delineate the interquartile range (IQR), with the middle thick segment indicating the sample median. Whiskers extend to the most extreme values within 1.5 × IQR. Data beyond this range are not reported. The sample size (n) for each boxplot corresponds to the number of samples reported in Table 1. Results of permutational analysis of variance (PERMANOVA) performed for each data type are shown within (**a**) and (**b**). The significance of the clustering was calculated for both the study of origin (16S: df = 16; F = 49; p-value = 0.0005; SMG: df = 6; F = 18.2; p-value = 0.0005) and the disease status (16S: df = 1; F = 31.4; p-value = 0.0005; SMG: df = 1; F = 7.9; p-value = 0.0005).

structure of the microbiome through a well-powered beta diversity analysis. Consistent with our previous report[12], samples did not cluster according to disease status (Fig. 1a, b). Even after removing the effect of the study of origin, only a weak separation was observed between PD and controls (Fig. 1c, d). Permutational multivariate analysis of variance (PERMANOVA) indicated that the disease status explains ≤1% of the variance in microbiome composition across studies, despite being statistically significant (Fig. 1). The study of origin instead explains a considerably higher proportion of variance, 19.9% and 7.7% for the 16S and SMG data, respectively, which is substantially higher

than those explained by the geographical origin of the studies (Supplementary Data 1; see also Supplementary Fig. 1 confirming the strong differences between studies). This highlights the high variability in microbiome composition across studies that is often observed in microbiome meta-analyses[12,13].

## Comparison of machine learning approaches

To assess how well the microbiome profiles could distinguish between control and PD samples, we first applied ML models to each dataset individually. We initially explored different filtering strategies, normalisation approaches, and ML algorithms implemented in the R package SIAMCAT[23]. Accuracies of ML models were evaluated using the area under the receiver operating characteristics curve (AUC). These comparisons were performed for the taxonomies of both 16S and SMG data. In general, for both types of data, retaining taxa detected in at least 5% of the samples in ten 16S and two SMG datasets resulted in profiles which allowed to build the most accurate ML models (Supplementary Figs. 2, 3). However, the accuracies of models varied substantially across ML algorithms and filtering/normalisation strategies (Supplementary Figs. 2, 3). For the 16S data, Random Forest classifiers performed in general better than the other algorithms tested, reaching a maximum AUC of $\geq 95\%$, observed in within-study cross-validation (CV) performed for the data of Zhang et al.[24] and Tan et al.[25] (Supplementary Fig. 4). For the SMG data, the Ridge regression and LASSO algorithm (LibLinear implementation) yielded the most accurate models with $\geq 85\%$ AUCs obtained for the study of Bedarf et al.[18] and Qian et al.[17] (Supplementary Fig. 5). For the sake of clarity and comparability, all results subsequently presented in the main text were obtained using the Ridge regression classifiers for both 16S and SMG data (Fig. 2 and Supplementary Fig. 6). Between the two data types, ML models built on SMG data had a higher average AUC for the within-study CV (also when compared directly on matched SMG and 16S data generated from the same samples in the study by Jo et al.[19] with AUCs of 82.4% and 70.4% for SMG and 16S data, respectively; Supplementary Fig. 7c) and a considerably lower variation compared to the 16S-based models (Fig. 2; SMG = 78.3% ± 6.5, 16S = 72.3% ± 11.7;

$t$ test: $t = 1.6$, $df = 19.6$, $p$-value = 0.13, effect size = $-0.6$, 95% confidence interval = $-1.9 - 0.2$). For both data types no correlation was observed across the studies included in this meta-analysis between the number of samples used to train the models and classification accuracies indicating that study heterogeneity overrides the expected gain in AUC with a higher sample size that is typical for ML applications to a single homogeneous dataset (Pearson correlation, 16S: $p$-value = 0.8; SMG: $p$-value = 0.95; Supplementary Fig. 7).

## Cross-study portability of the ML models

Given that study-specific PD models in many cases showed promising accuracies, we next assessed generalisation across studies, i.e., examined their prediction accuracies when tested on all other data sets. We performed study-to-study validation (cross-study validation; CSV) for both 16S and SMG data, by treating all data sets a study-specific model had not been trained on as independent test sets. Compared to the performance estimated through within-study CV, CSV performances were significantly lower for both SMG and 16S data ($t$ test: SMG, $t = -5.2$, $df = 8.8$, $p$-value < 0.001, effect size = $-2$, 95% confidence interval = $-3.6$ to $-1$; 16S, $t = -4.2$, $df = 17.4$, $p$-value < 0.001, effect size = $-1.1$, 95% confidence interval = $-1.7$ to $-0.7$; Fig. 2 and Supplementary Figs. 4–6). In general, 16S datasets showing high AUC in the within-study CV (i.e., Zhang et al.,[24] and Tan et al.[25]) could also be better classified by external models (i.e., models built on other datasets; Supplementary Fig. 4). However, the models trained on these datasets showed a much lower performance when tested across other studies (Supplementary Fig. 4). A similar pattern was observed for the SMG data (Fig. 2 and Supplementary Fig. 5). For example, the datasets of Bedarf et al.[18] resulted in a model with an AUC of 85% in the within-study CV, but an average AUC of only 57.4% in CSV (Supplementary Fig. 5). Low model generalisation evident from low CSV accuracies could neither be explained by differences in age and sex distribution between test and training set (Supplementary Fig. 8; coefficient $p$-values in linear models > 0.05), nor by the geographic origin of the samples (Western vs Eastern countries; Supplementary Fig. 8; one-way ANOVA $p$-values > 0.05). CSV AUCs obtained for the SMG models were

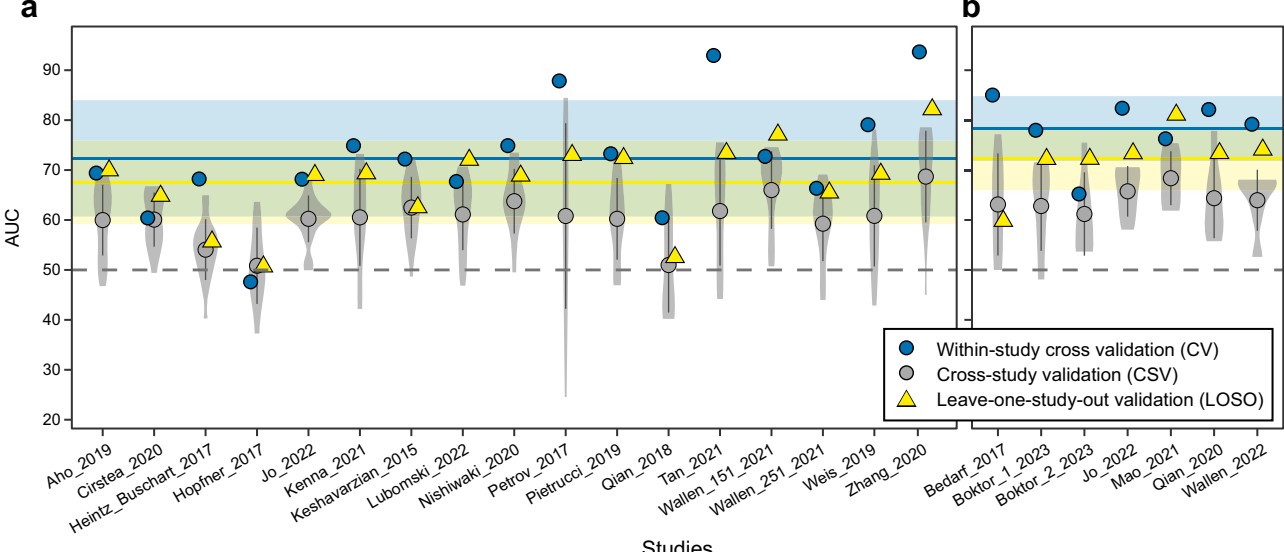

**Fig. 2 | Performance of the microbiome-based ML models.** AUC values of the ML models across 16S (**a**) and SMG (**b**) datasets. Violin plots depict the distribution of ML performances (AUCs) calculated for cross-study validation (CSV; where the study-specific models were tested on data from every other study with AUCs reported for each test set along the $x$-axes; each summarising 16 and 6 AUCs for 16S and SMG data, respectively). Within the violin plots the average AUC and the standard deviation are reported (grey dots and vertical lines). Blue circles and

yellow triangles indicate the AUCs for within-study cross-validation (CV) and leave-one-study-out validation (LOSO), respectively. LOSO AUCs were obtained by training ML models on all but one dataset, which was then used to evaluate the model as a hold-out set (along the $x$-axis). The horizontal shaded areas and lines indicate the standard deviation and average of the AUCs, respectively, for LOSO (yellow) and CV (blue). The grey dashed line marks the 50% AUC threshold indicating random guessing.

higher than those for the 16S models (t test: $t = -3.2$, $df = 65.1$, p-value = 0.002, effect size = −0.5, 95% confidence interval = −0.8 to −0.2; average AUC SMG 64.2% ± 7.4, average AUC 16S 60.1% ± 9.73). No clear correlation was observed between the number of samples used for training and the CSV classification accuracies (Pearson correlation, 16S: p-value = 0.9; SMG: p-value = 0.22; Supplementary Fig. 7). These low overall CSV performances indicate large inter-study variability in microbiome composition, consistent with the above-mentioned PERMANOVA results (Fig. 1 and Supplementary Fig. 1).

The 16S datasets varied greatly in sequencing depths (Supplementary Fig. 9), which may negatively affect the generalisation capabilities of ML models. To test this possibility, we built ML models on rarefied data (see "Methods"), and compared their accuracy in CSV to those of the not-rarefied models. For the majority of the ML algorithms, AUCs did not change significantly between the two approaches (average difference in AUCs < 0.7%; Supplementary Fig. 10). Only Ridge regression was sensitive to heterogeneity in sequencing depth, although AUCs differences were minimal (average difference in AUCs ≤ 1.3%; Supplementary Fig. 10). Another factor potentially affecting CSV performances is study-associated heterogeneity in microbiome profiles – due to technical or biological differences (here study effects, elsewhere also referred to as batch effects). As study effects appeared considerably stronger in the 16S data than the SMG data (Fig. 1), we investigated if correcting for study effects in the 16S data would increase the overall model portability. We explored various available batch correction approaches to reduce study heterogeneity while ensuring that all methods were blind to the labels (PD vs controls) to avoid over-optimistic evaluations[26]. However, none of the batch correction approaches used here significantly increased the average AUC in the CSV evaluations (Supplementary Fig. 11 and Supplementary Data 2). This suggests that currently, available batch correction methods may be of limited practical value for improving the cross-study portability of microbiome classifiers.

Next, we examined whether model performance could be improved by pooling data across studies in comparison to models trained on single studies. This can be assessed using leave-one-study-out (LOSO) validation, in which all data are combined except for the data from one study that is used to evaluate the model. For both 16S and SMG data, we observed LOSO model performances to be significantly better than for CSV (Fig. 2; t test: 16S, $t = -3.5$, df = 18.8, p-value = 0.002, effect size = −0.8, 95% confidence interval = −1.4 to −0.4; SMG, t = −3.1, df = 9, p-value = 0.01, effect size = −1.2, 95% confidence interval = −2.2 to −0.3), even though there was still considerable variability across held-out studies. Between data types, average LOSO AUCs for SMG were higher than those obtained for the 16S data (t test: $t = -1.5$, df = 14.9, p-value = 0.15, effect size = −0.6, 95% confidence interval = −1.7 - 0.1; LOSO AUC average: SMG = 72.3 ± 6.3, 16S = 67.5 ± 8.3). To examine additional factors with a probable influence on LOSO AUCs, we next investigated the composition of the training data. For this, we performed variations of LOSO validations (one for each SMG hold-out dataset) in which the training sets were constructed by progressively increasing the number of pooled studies from 2 to 6 for all possible combinations (57 models per test set for a total of 399 models). The results of this analysis show a dependence of the resulting AUC on the test set, which explains a considerable proportion of the variance in AUC (intraclass correlation coefficient = 0.19; Supplementary Fig. 12). Nevertheless, LOSO AUCs also did increase significantly with an increasing number of training samples (coefficient p-values in linear models < 0.01; 15% of variance explained; Supplementary Fig. 12). Furthermore, we hypothesised that models built on data collected within a similar population (i.e., studies from the same continent) might be more similar to each other, corresponding to higher CSV and LOSO performances. To assess this, we extracted the feature coefficients from all models built on the 16S and SMG data and visualised their similarity using ordinations (Canberra distances;

Supplementary Fig. 13). However, we observed only a weak separation by continent of origin, which was statistically significant only for SMG data (PERMANOVA p-values: 16S = 0.40; SMG = 0.04). These results are consistent with the lack of association between the Western/Eastern origin of the study and the CSV AUCs (Supplementary Fig. 8), suggesting that model accuracy in generalising across studies is not primarily limited by geographic differences, but rather by other study-specific characteristics.

In light of the large variability of cross-study generalisation accuracies of PD models, we asked if a universal subset of features exists that is sufficient to robustly discriminate PD from controls. As the SMG data resulted in more generalizable models, we performed another set of LOSO validations in which we built models only on the 20 features with the strongest difference in abundance between PD and controls in each training set (to avoid over-optimistic performance estimation, we did not select features globally across all SMG datasets). These reduced models generally exhibited slightly decreased LOSO accuracies (Fig. 3a). However, for the test sets of Bedarf et al.[18] and Boktor et al.[16] (for the dataset named Boktor_1 here) this approach considerably increased AUCs, resulting in an overall average LOSO AUC almost identical to the one obtained using the full features set (72.3% vs 72.4%). These results indicate that even when based on a concise gut microbial signature, models can classify PD vs. controls with reasonable accuracy when trained on pooled data from multiple studies and evaluated on held-out populations. In general, the taxa selected were consistent across training sets (Fig. 3b). Exceptions from this were mostly taxa (not) selected exclusively when Wallen et al. was used as a test set, which is likely due to the fact that this dataset is considerably larger than the others ($N = 724$ versus an average $N = 100$ for the others) and thus has a larger influence on the statistical analyses. However, six taxa were selected in all of the seven training sets, suggesting consistent associations of these bacteria with PD.

## Cross-disease prediction

A final important aspect of externally validating PD models is addressing their disease-specificity, that is to check to what extent they wrongly predict patients affected by neurodegenerative diseases other than PD. To assess cross-disease prediction rates, we tested PD models on data obtained from studies investigating other neurodegenerative diseases. We performed this cross-disease validation using only 16S data due to a scarcity of publicly available SMG data for such diseases. We tested models built for each PD 16S dataset on data from Alzheimer's disease (AD) and Multiple Sclerosis (MS). The observed cross-prediction rates (assessed in terms of the false positive rate, FPR, on AD and MS samples and compared to the PD-internal FPR of 10%) varied greatly across the PD-specific ML models, ranging from 0% to almost 100%, with an average of 35.1% (Fig. 4). However, cross-disease prediction drastically improved when using LOSO models, as their average FPR was reduced to 18.7%, which is only moderately higher than the expected 10% FPR for PD-internal control groups. Our finding that disease specificity of ML models can be significantly improved by training on data pooled across multiple studies confirms earlier reports on the effectiveness of this approach[23].

## Comparison between taxonomic and functional microbiome profiles

Not only taxonomic profiles but also functional microbiome profiles, derived from SMG data to capture a broad range of metabolic and other pathways, have previously been used for building classification models. Here we specifically used KEGG orthologous groups (KOs), KEGG modules, KEGG pathways, gut metabolic modules (GMMs[27]), or gut-brain axis modules (GBMs[28]; manually curated microbial metabolic pathways of relevance for gut health and gut-brain axis) to investigate whether models based on functional or taxonomic profiles yield better

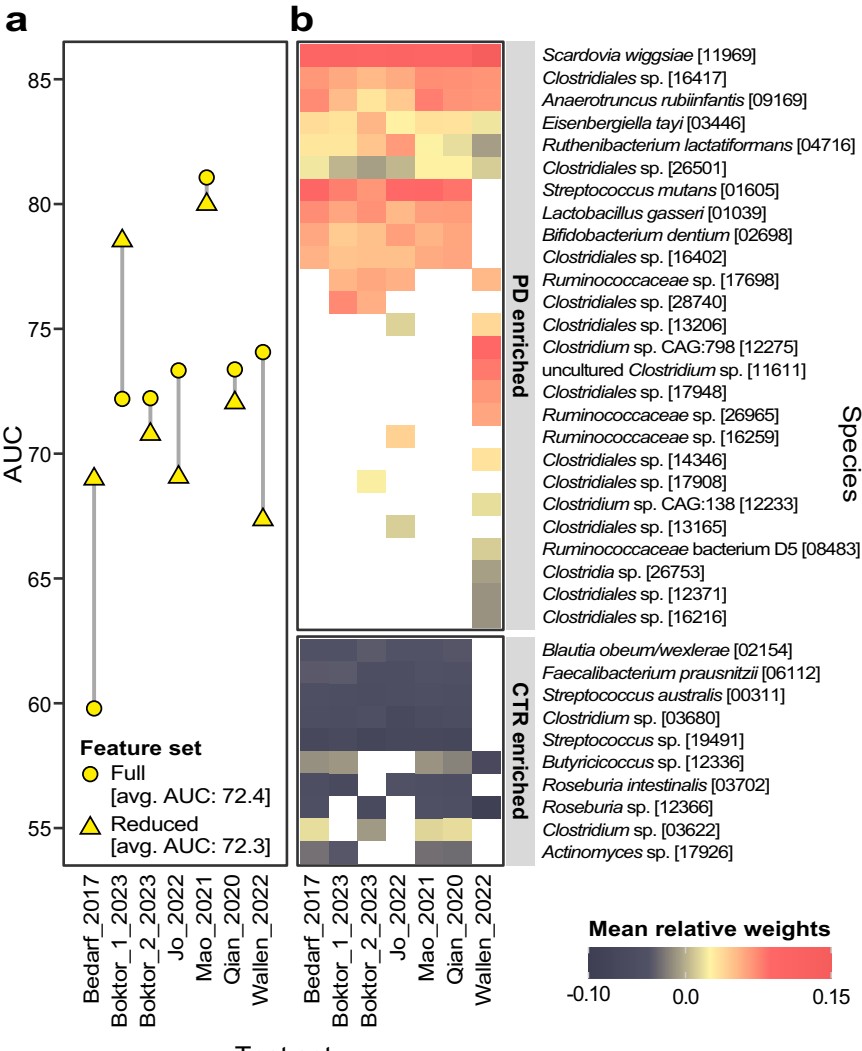

**Fig. 3 | Classification accuracies are maintained when only a subset of species are used.** For each of the 7 training sets used for LOSO validation within the SMG datasets, we conducted a feature selection based on a differential abundance analysis performed on the training set. We then selected the 20 features with the largest absolute effect size and used them to build new Ridge regression models, which were then tested on the held-out data. **a** LOSO model accuracies (avg. -average). **b** Heatmap of the joint set of features selected in at least one dataset with average relative weights colour-coded. Here, features are divided based on their enrichment in PD or controls. In both panels the models are referred to by the study used as a test set (*x*-axis). White tiles in the heatmap refer to species that were not included in the respective model.

accuracy[13,17]. We found that models based on functional profiles perform in general slightly worse than those built on taxonomic profiles (although differences were in most cases not statistically significant, Fig. 5 and Supplementary Data 3), and this was consistent across the different types of functional profiles examined (Fig. 5, Supplementary Figs. 14–18 and Supplementary Data 3). The only exception was observed in the CSV, where the average AUC obtained for the KO profiles was slightly higher than those obtained for species ($64.8 \pm 8.4$ vs $64.2 \pm 7.4$, Supplementary Data 3). Among the different functional profiles, KOs performed best in discriminating PD from controls and had the highest CSV performances compared to KEGG modules and pathways, GMMs, or GBMs (Fig. 5; Supplementary Fig. 14–18 and Supplementary Data 3). Similar to the taxonomic profiles, across the different types of functional profiles CSV accuracies were considerably lower than those obtained for the within-studies CV (Fig. 5 and Supplementary Figs. 14–18). These results indicate that the use of functional profiles does not significantly improve classification accuracies or ML model portability (as assessed by CSV) when compared to taxonomic profiles.

## Taxa associated with PD

To systematically identify taxa consistently associated with PD across datasets, we performed a meta-analysis on relative abundances of gut microbial taxa (Fig. 6 and Supplementary Datas 4, 5). This was done by calculating Generalised Odds Ratios (Gen. Odds), pooling the effect sizes using random effect meta-analysis, and correcting *p-values* for multiple testing using the Benjamini-Hochberg method (FDR). We found that taxa within the *Lachnospiraceae* family belonging to the genera *Roseburia*, *Blautia*, and *Fusicatenibacter* were strongly depleted in the microbiome of PD patients in both 16S and SMG data. Similarly, we detected the genus *Agathobacter*, within the *Lachnospiraceae* family, to have a strongly reduced abundance in PD in the 16S datasets. Affiliated to this genus are uncharacterised species (mOTUs 03657 and 12366) which were depleted in PD patients across the SMG datasets. Similarly, *Faecalibacterium* (family *Ruminococcaceae*) was found strongly and consistently depleted in PD. The high-resolution profiling we performed for the SMG data allowed us to identify multiple species within the *Faecalibacterium* genus and multiple strains within the *Faecalibacterium prausnitzii* species (mOTUs 06112, 06110, 06109,

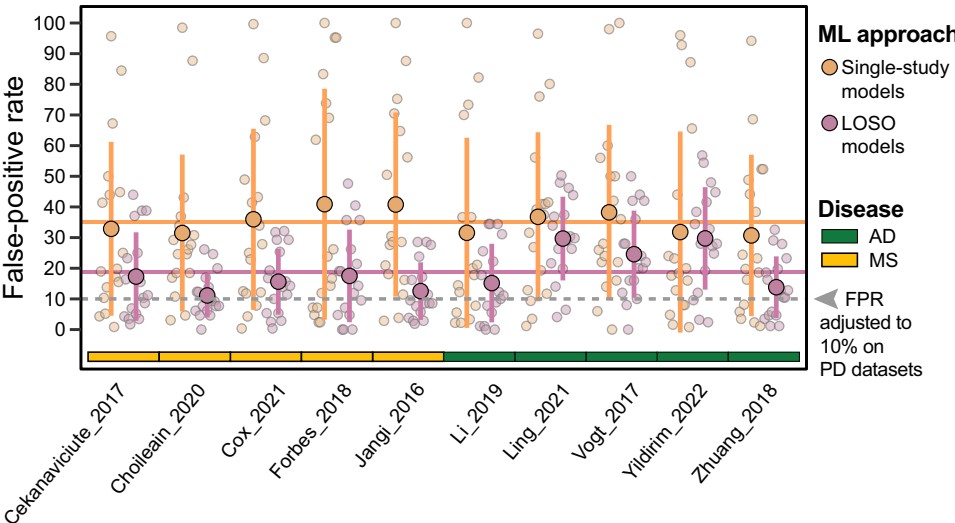

**Fig. 4 | Disease specificity of classification models is significantly improved by pooling data from multiple studies.** Dots indicate the proportions of samples spuriously predicted as PD in datasets of other neurodegenerative diseases (to assess the disease specificity of PD classification models). All classification models (Ridge regression) trained on individual PD studies (16S data only) as well as on pooled data (LOSO) were evaluated for false-positive prediction (FPR) rates on datasets obtained for other diseases (AD = Alzheimer's disease; MS = multiple sclerosis, all data generated using 16S sequencing). Models were originally adjusted to a 10% FPR on the controls from PD datasets. Thus, an FPR above this level in patients with other diseases indicates a lack of disease specificity as previously established[13]. Average and standard deviation of AUC values (17 in total) are reported for models trained on a single study (orange dots and vertical lines) as well as for LOSO models (purple). On average, LOSO models were found to be much more disease-specific (average FPR = 18.7%, horizontal purple line) than models trained on a single dataset (average FPR = 35.1%, horizontal orange line).

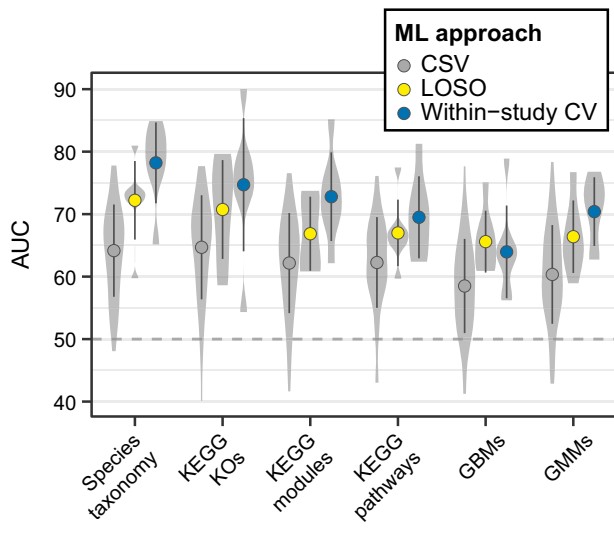

**Fig. 5 | Taxonomic profiles perform better than functional profiles in discriminating PD from controls.** Violin plots depict the density of AUC values obtained across ML approaches (within-study cross-validation, CV; study-to-study validation, CSV; and leave-one-study-out validation, LOSO). Within the violin plots the average AUC and the standard deviation (circles and vertical lines) are reported (based on 42 AUCs for CSV and 7 AUCs for CV and LOSO). Performances for ML models (Ridge regression classifiers) built on taxonomic (mOTUs) or various functional profiles are compared as labelled along the *x*-axis. The horizontal grey dashed line marks the 50% AUC threshold indicating random guessing. KO KEGG orthologous gene families, GMMs gut metabolic modules, GBMs gut-brain axis modules, species taxonomy mOTUs_v3 species profiles.

06108) depleted in the microbiome of PD patients. The species showing the strongest depletion in PD metagenomes belonged to the *Butyricicoccus* genus (Fig. 6 and Supplementary Data 5). However, this association was not detectable in 16S datasets, even though the corresponding family *Butyricicoccaceae* was reported as PD-depleted in our previous meta-analysis[12].

For many taxa enriched in PD, we observed similarly good concordance between 16S and SMG data. For example, the five genera with the strongest enrichment in the 16S datasets had related mOTUs enriched in the SMG data (Fig. 6 and Supplementary Datas 4, 5). Differently from previous studies[12,15,17,29], we detected in both 16S and SMG data the *Ruthenibacterium* genus and the *Ruthenibacterium lactatiformans* species as the most enriched taxa in the PD gut microbiome. In addition, taxa within the genera *Alistipes, Anaerotruncus, Enterococcus, Porphyromonas, Scatomorpha, Limiplasma, Bifidobacterium, Christensenella, Streptococcus* were all consistently enriched in the 16S and the SMG datasets. However, we also observed several differences in the taxa associated with PD between the two sequencing methods (Supplementary Datas 4, 5). For example, in SMG data we detected the potential pathogenic species *Turicibacter sanguinis* (mOTU 04703), and multiple species within the order Clostridiales enriched in PD samples, but the respective genera were not significantly PD enriched in the 16S datasets. Taxa within the *Lactobacillus* genus were enriched as well in PD samples. Recently, this genus has been reassessed taxonomically and several new genera have been created[30]. As we used an up-to-date version of the Genome Taxonomy Database (GTDB v207) to obtain high-resolution taxonomic classification of 16S-derived ASVs, we could identify the genera *Limosilactobacillus, Lactobacillus, Lacticaseibacillus*, and *Ligilactobacillus*, within the *Lactobacillus* sensu-lato, all enriched in the PD gut microbiome (Fig. 6 and Supplementary Data 4).

Despite significant differential abundance detected in the pooled meta-analysis, many taxa exhibited significant abundance shifts in only a fraction of the individual datasets, in agreement with previous findings[12,16] (Fig. 6b and Supplementary Data 4, 5). This most likely reflects both the variability observed across studies and the low statistical power in smaller datasets we re-analysed. To more deeply investigate the robustness of PD associated taxa, we assessed if these might be confounded by sex, age, or medication usage. For this we re-analysed datasets with available metadata using linear models in which

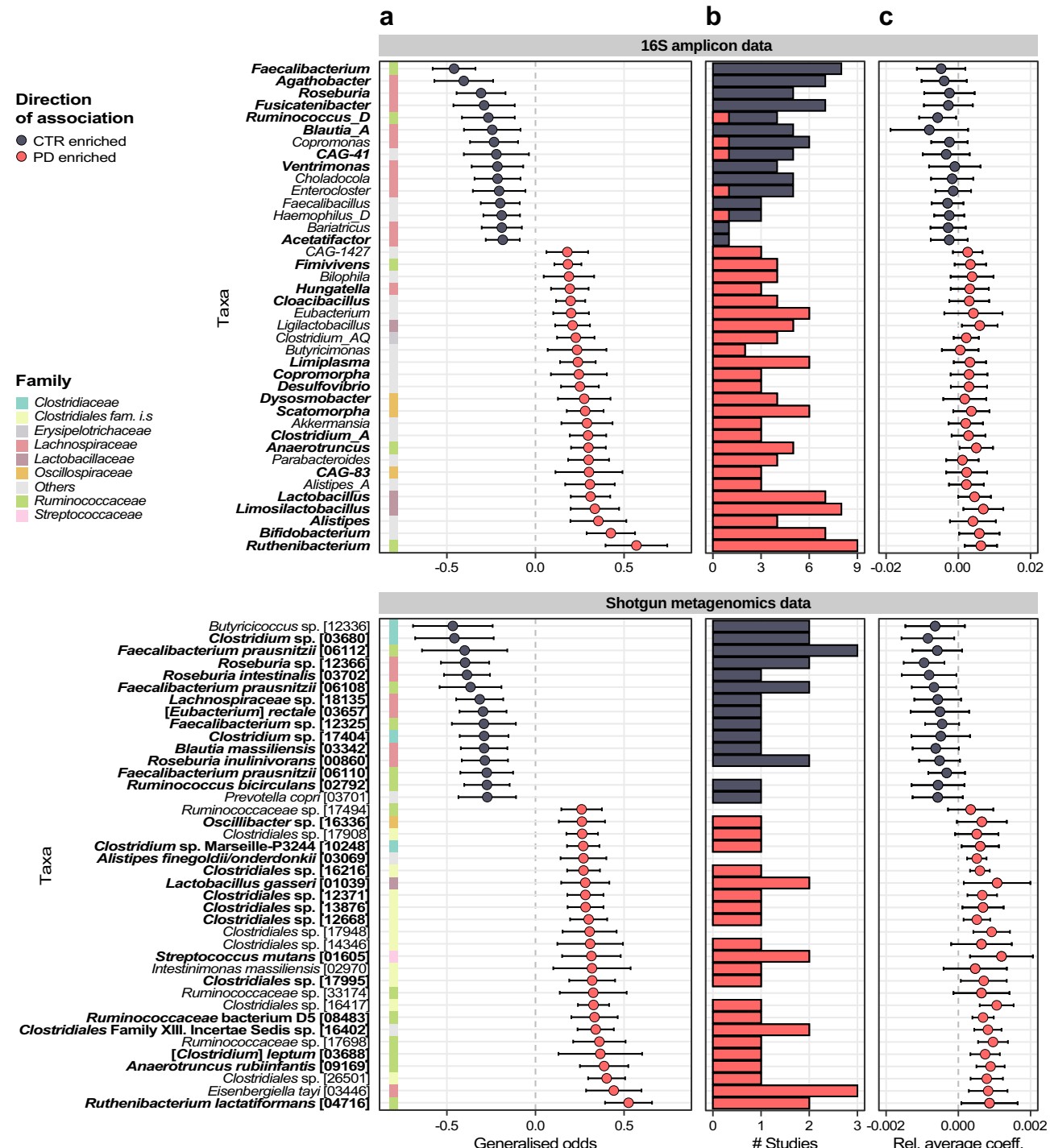

**Fig. 6 | Taxa showing significant differences in abundance between PD and controls (CTR). a** Univariate differential abundance testing was performed independently for each dataset using Agresti Generalised Odds ratios on the taxa obtained from 16S and SMG profiles. Results (Generalised odds) were pooled using random effect meta-analysis and displayed together with 95% confidence intervals. Tests were performed using all samples (see Table 1) of the 17 16S and 7 SMG datasets (see Methods for details). **b** Number of individual studies in which each taxon significantly differs in abundance between PD and controls. **c** Average Ridge regression coefficients and standard deviations calculated from all models built on each dataset (17 and 7 models for 16S and SMG data, respectively). Only 40 taxa with the largest effect sizes are shown. Genera and mOTUs with similar directions of enrichment across 16S and SMG data are displayed in bold.

these covariates were accounted for. This analysis indicated that only a minor fraction of the taxa associated with PD (< 23%) were potentially confounded by sex, age, or medication usage, and in general the taxa with the strongest abundance shift were not affected by covariates (Supplementary Datas 6–9). When comparing the results of differential abundance tests applied to each taxon with their influence in the

classification models (i.e., their coefficient size in the Ridge regression classifiers), we found these to be remarkably consistent (Supplementary Fig. 19). The sign of model coefficients for these taxa mostly matched the direction of association from univariate analysis although variability in the average Ridge regression weights across datasets was evident (Fig. 6c and Supplementary Datas 4, 5).

## Gut microbial gene functions associated with PD

To explore changes in gut microbial functionalities in PD patients relative to controls, we extended the differential abundance meta-analysis to microbial genes and pathways as defined by the KEGG orthologous groups (KO), KEGG modules, KEGG pathways, GMMs, and GBMs (Fig. 7 and Supplementary Datas 10–14). We complemented this approach with an enrichment analysis to detect those pathways that were significantly enriched in KOs showing differential abundances between PD and controls (Supplementary Data 15). Below we highlight those gut microbial functions with a possible relation to Parkinson's aetiology or symptomatology. Modules related to the degradation of complex polysaccharides and sugars were strongly depleted in PD, in agreement with previous reports[15,18] (KEGG modules: M00631, M00061, M00081; GMM: MF0001, MF0003, MF0004, MF0022, MF0010, MF0018, MF0002; KEGG pathways enriched in KOs depleted in PD: ko00040, ko00520; Supplementary Datas 11, 13, 15). In contrast to these results, some functionalities related to the production of propionate and butyrate were enriched in PD (MF0093, MF0094, MF0089, Supplementary Data 13).

Several pathways involved in the metabolism of amino acids showed a significant difference in abundance between PD and controls (Fig. 7a and Supplementary Datas 10–15). These pathways are relevant in the context of the gut-brain axis, as amino acids, especially tryptophan and tyrosine, are precursors of neurotransmitters that have altered concentration in PD[31]. Our results suggest that the PD gut microbiome has an increased ability to degrade tryptophan as genes encoding enzymes involved in this process were significantly enriched in PD gut metagenomes, while those involved in tryptophan synthesis were depleted (KEGG pathway: ko00380, KEGG Module: M00038, GBM: MGB049, MGB004, MGB005 and GMM MF0025; Supplementary Datas 10–14). Our data also hint at an increase in microbial tyrosine turnover in the gut of PD patients, as we detected a significantly higher abundance of genes for both tyrosine degradation and synthesis (KEGG pathway: ko00350, KEGG modules: M00044, M00042, and M00040, and the GMM MF0027; Supplementary Datas 10–14). Within this pathway, the gene coding for tyrosine decarboxylases TyrDC (K22330) was enriched in PD gut metagenomes. Intriguingly, this enzyme also catalyses the degradation of the main PD medication, L-dopa, in *Lactobacillus* sp. and *Enterococcus* sp[32]., suggesting that PD medication regimes might influence the metabolism of the PD gut microbiome. Indeed, our analysis suggested that some of these pathways might be affected by PD-related and non-PD-related medications, but not L-dopa (Supplementary Datas 6–8 and Supplementary Fig. 20). Significantly altered abundances were also observed for genes related

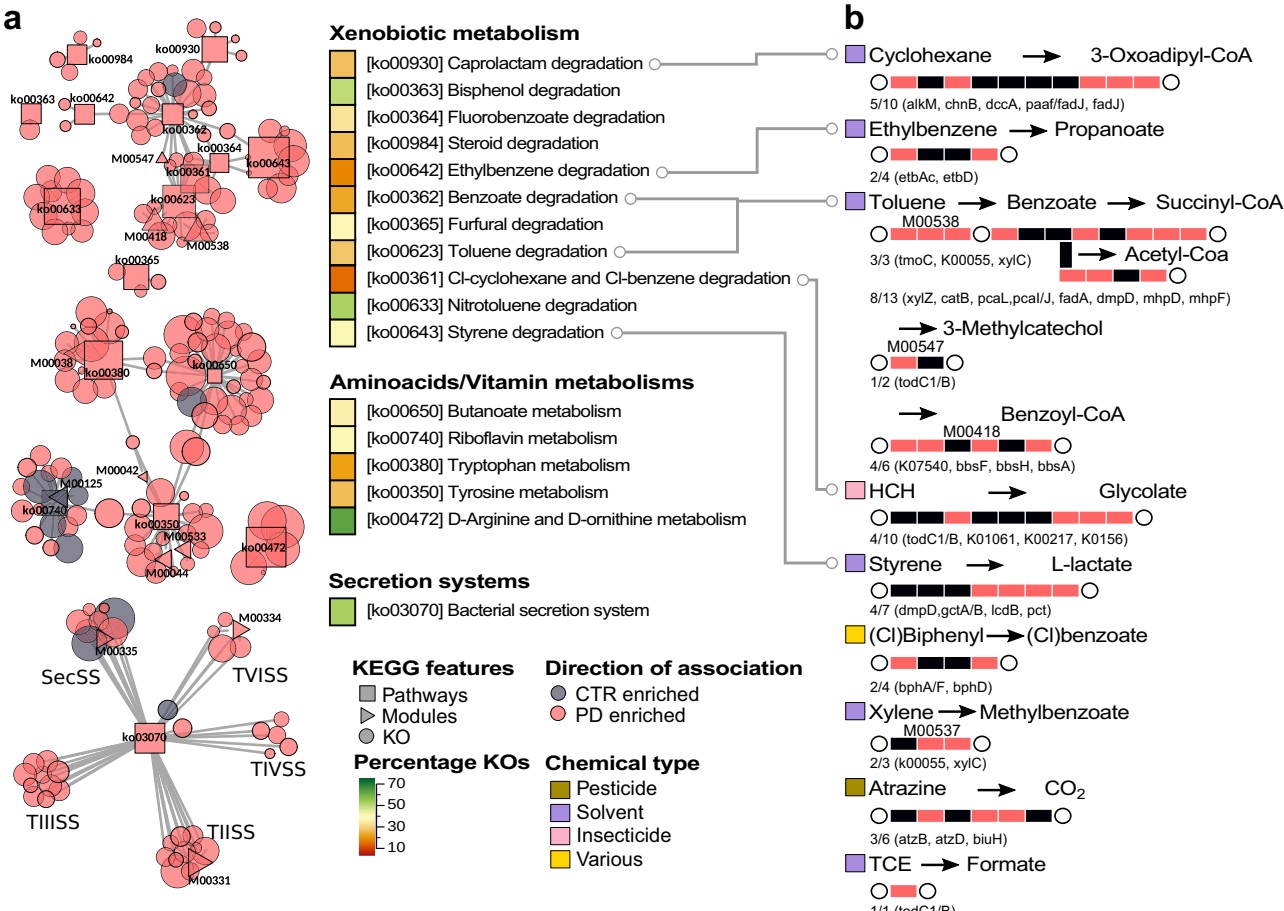

**Fig. 7 | KEGG functionalities significantly differ in abundance between PD and controls (CTR). a** Selected KEGG-based functionalities are displayed using a network in which the relationships between pathways, modules, and KO are indicated. The type of functionalities and the direction of enrichment are displayed using shapes and colours, respectively (see key). The size of each node in the network is proportional to the effect size (larger effect size = stronger enrichment). Only pathway and module nodes are labelled. Heatmap stripes indicate the proportion of differentially abundant KOs, out of all KOs in the pathway. TIISS = type II secretion system; TIIISS = type III secretion system; TIVSS = type IV secretion system; TVISS = type VI secretion system; SecSS = Sec secretion system. **b** For the xenobiotic metabolisms, a summary of the most representative reactions catalysed by KOs enriched in PD is shown. Each rectangle corresponds to an enzyme or a complex of enzymes, representing a metabolic reaction within the respective KEGG pathway. Rectangles are depicted in red when at least one of the KOs catalysing the enzymatic reaction was detected as enriched in PD, and in black when not. The number of reactions with KOs enriched in PD and the respective KO gene names are reported below each graph. Circles depict metabolites and xenobiotics, which are colour-coded to reflect their use by humans (see key).

to the metabolism of glutamine, glutamate, and 4-aminobutyrate (GABA), which are all essential amino acids for brain metabolism and function. PD metagenomes were depleted in genes encoding enzymes for glutamate synthesis and showed enrichment in genes involved in its degradation (enriched in PD: equivalent modules MF0032 and MGB051; depleted in PD: equivalent modules MGB007 and MF0047; KEGG KOs: K01846, K19268, K04835, K00265, K00266, K00284; Supplementary Data 10, 13, 14). Finally, enzymes catalysing the degradation of GABA and gamma-Hydroxybutyric acid, a natural GABA precursor, as well as the last step of the glutamate conversion into succinate through GABA, showed significantly higher abundance in PD metagenomes (KEGG module: M00027; KEGG KOs: K00135; GMM MF0076 and GBM MGB039; Supplementary Datas 10, 11, 13, 14). Although these data are consistent with the increased ability of the PD microbiome to degrade GABA, we also detected an enrichment, albeit with smaller abundance shifts, of enzymes catalysing GABA synthesis (GBM: MGB021 and MGB020; KEGG module: M00136) and a potential confounding effect of age on the abundances of these functions (Supplementary Data 9). Together this suggests complex microbiome influence on both production and degradation of GABA.

PD gut metagenomes were enriched in genes encoding proteins involved in the adhesion to, interaction with, and manipulation of host cells, as well as the resistance against host immune responses. Specifically, the KEGG pathway for bacterial secretion systems was significantly more abundant in PD metagenomes (ko03070; Fig. 7a and Supplementary Data 12). Secretion systems are complex molecular machineries used by bacteria to release effector proteins in the surrounding environment or into neighbouring cells[33]. The type III, IV, and VI secretion systems are used by pathogens to inject effector molecules into host cells to manipulate their defence and immune systems[34]. Within this pathway, we observed 52.7% of KOs to be differentially abundant between PD and controls, with the types II, III, IV, and VI secretion systems showing the clearest enrichment in PD. Additional KOs related to the type VI secretion system were enriched in PD as well (K11890, K11895-7, K11900, K11909, K12210-1, K12213, K12217-8; Supplementary Data 10). Similarly enriched were several modules and KOs involved in bacterial resistance against cationic antimicrobial peptides (CAMPs; KEGG modules: M00730, M00739, M00744, M00723, M00722, M00726, M00725; Supplementary Data 11 and 15). CAMPs are important host defence mechanisms produced at sites of infection and/or inflammation[35]. Hence, finding an enrichment of these defence mechanisms suggests an ongoing host immune response towards microbes. Some of the above associations might be potentially confounded by sex and age (Supplementary Data 9 and Supplementary Fig. 20), as these functions showed higher abundances in the older population and in males, which are both, however, known intrinsic risk factors for PD[1,4]. Another way in which bacteria can interact with their host is by producing extracellular structures called curli fibres that are involved in cell adhesion, biofilm formation, and bacterial virulence[36]. Confirming previous findings[15], KOs for curli fibres showed a significantly higher abundance in PD (K04337-8, K06214, K04334-5; Supplementary Data 10). These amyloid-like bacterial proteins have attracted considerable interest in relation to PD as they have been shown to promote αSyn aggregation and motor impairment in mice[37,38]. Altogether these results indicate an enrichment of potential pathogenic functions in the gut microbiome of PD patients and suggest an increased activation of host defence mechanisms towards infectious agents.

Finally, our analyses of gut microbial functions revealed that multiple pathways within the KEGG class *"Xenobiotics biodegradation and metabolism"* were significantly enriched in PD. While xenobiotic metabolism has not been thoroughly investigated in PD metagenomes previously, it is highly relevant since exposure to environmental xenobiotics (e.g., pesticides, herbicides, solvents) is one of the main non-genetic risk factors for developing PD[4,39–42]. In these enriched

pathways, between 15.4 and 52.6% of all KOs were significantly more abundant in PD (Fig. 7a). Some of these KOs (e.g., K04072, K00121, K00170) can be part of the central metabolism and hence might not necessarily be involved in xenobiotic degradation. Moreover, it cannot be excluded that these enzymes may metabolise medications taken by PD patients. Indeed, we detected a minority of these features to be potentially confounded by medication intake in addition to sex and age (Supplementary Datas 6–9 and Supplementary Fig. 20). However, we detected many unconfounded KOs enriched in PD that are specifically involved in the metabolism of environmental xenobiotics: for example, the 2-haloacid dehalogenase K01560, which takes part in the degradation of halogenated hydrocarbons (Supplementary Data 10). These molecules have been widely used as solvents, industrial chemicals, pesticides, and herbicides, and have been linked with PD before[39,40]. For example, recent epidemiological studies suggested that individuals exposed to water contaminated with trichloroethylene (also known as trichloroethene or TCE) had a 70% increased risk of developing PD[40]. Interestingly, our analysis revealed that PD gut metagenomes were enriched in K03268 and K18089, which encode enzymes that can catalyse the conversion of TCE into formate. Considering the relevance of xenobiotics in PD aetiology, we further inspected the PD-enriched KOs manually to identify those involved in xenobiotic metabolism and related pathways (Fig. 7b). In addition to the KOs belonging to PD-enriched pathways, we observed a significant increase in abundance of other KOs involved in xenobiotic metabolism, even though the whole pathway did not pass our significance threshold. For example, PD metagenomes had a higher abundance of the genes *atzB*, *atzD*, and *biuH* (K03382, K03383, K19837; Supplementary Data 10), which encode enzymes that catalyse the degradation of atrazine, which is a widely used chlorinated pesticide that showed dopaminergic toxicity in rat models[41]. In summary, our meta-analysis revealed compelling associations between the microbiome functionalities and PD matching known risk factors for the disease.

## Discussion

In recent years, several studies have suggested that the gut microbiome might be leveraged to support PD diagnosis[17–20,22]. However, a consensus on gut microbiome features associated with PD and a thorough evaluation of microbiome-derived biomarkers for PD is lacking. The extensive ML validation and optimisation we performed here underline that within most study populations gut microbiome-based ML models can accurately discriminate PD from controls. However, ML models were generally study-specific, i.e., poorly generalised to data from other studies (cross-study portability tested using CSV). This may reflect the fact that PD is a very heterogeneous disease in terms of aetiology, pathophysiology, and symptomatology[6,43]. This biological heterogeneity is rarely accounted for in microbiome studies, as samples come from patients (i) affected by different PD subtypes; (ii) having different PD severity resulting in different lifestyles; (iii) having individual medication regimes; (iv) reporting disparate histories of medical conditions and exposure to risk factors (e.g., xenobiotics). All these aspects might exert heterogeneous influences on microbiome composition, and contribute to the low study-to-study portability and high variability in accuracy of ML models across datasets observed here. Moreover, these differences can potentially confound the associations between microbiome features and disease conditions. To thoroughly assess potential confounders, the scarcity of standardised publicly available metadata poses a severe limitation. Nonetheless, here we did analyse the metadata available for some studies, which suggested that a minor part of PD-associated microbiome features may be confounded (< 28.2%). However, it is worth noting that to identify all potential confounding effects, larger datasets with standardised metadata are required. Hence, caution is needed in concluding on the value of gut microbiome biomarkers for PD diagnostics. On a positive note, the data from Bedarf et al. [18] that comprises only male, L-dopa-

naive, early PD patients, allowed us to build ML models that classify PD cases with high accuracy. In their study, the authors ruled out an overall influence of PD medication on the microbiome abundance. Hence, it is reasonable to expect that the gut microbiome of these patients more closely resembles the one of undiagnosed/early PD patients. The results related to this particular study suggest that microbiome signatures may capture truly PD-associated signals. However, it is also evident that the models built for this study did not generalise well to other datasets, suggesting that this PD population is relatively dissimilar to those of the other studies we re-analysed. More generally, when pooling SMG data for model building, generalisability increased (see LOSO validations), and this was also true when only a subset of taxa was selected, indicating that these features could be truly associated with PD. Moving forward, the diagnostic potential of such gut microbiome biomarkers would need to be explored in larger multi-centre studies of drug-naive early PD patients, or of high-risk individuals, as has been recently attempted in two independent investigations[44,45]. Another important prerequisite for future clinical application of microbiome-derived biomarkers for PD is their disease specificity, i.e., their capability to distinguish the PD microbiome signature from those of other neurological diseases. Towards this aim, we demonstrate that pooling training data from multiple studies was also an effective strategy for building PD-specific ML models with a low propensity for making false-positive predictions on microbiome profiles from other neurodegenerative diseases.

Our large-scale meta-analysis further adds to a better understanding of PD processes to which the gut microbiome may contribute. First, we found the gut microbiome of PD patients to be depleted in bacteria known to ferment complex carbohydrates into SCFAs and in pathways involved in complex carbohydrate degradation. While this is in agreement with earlier studies[12,15,16,18,46], we report this depletion in the largest and most diverse dataset thus far analysed, which strongly suggests that it is a common feature in PD across patient populations. Low levels of SCFAs have been linked to compromised gut health, increased gut permeability and inflammation, as well as prolonged transit time, and have often been recorded in faeces of PD patients, who can suffer from compromised gut health[9,47–49]. However, the fact that some functions related to SCFA production were enriched in PD indicates that caution needs to be exerted in concluding on the metabolic output of the microbiome-based only on metagenomic data. Future multi-omics approaches (e.g., metagenomic, metaproteomics, and metabolomics) are required to clearly establish how the gut microbiome contributes to metabolite concentrations in the gut of PD patients. Our data indicate that another general feature of the PD gut microbiome is the enrichment of lactic-acid producing bacteria (e.g., *Lactobacillus* sensu-lato, *Bifidobacteirum*, and *Ruthenibacterium*), in agreement with earlier findings[12]. Some *Lactobacillus* strains encode the enzyme tyrosine decarboxylases TyrDC (K22330) which catalyse the degradation of the main PD medication L-dopa[32]. Here, we verified at a large scale that among the lactic acid-producing bacteria enriched in PD, TyrDC is only encoded by some taxa within the genus *Lactobacillus* sensu-lato and *Bifidobacterium* (Supplementary Data 16). Hence, the use of L-dopa alone does not explain the enrichment of these bacteria in PD, as their abundances were also not associated with PD medication usage (Supplementary Datas 6–8). Whereas lactic acid-producing bacteria are generally considered beneficial commensals, some of them have also been found enriched in other inflammatory conditions affecting the gut (i.e., IBD) and it has been suggested that they might take advantage of microbiome imbalance in a proinflammatory environment[50,51]. Further experimental work is required to clarify whether their increased abundances have an impact on the pathophysiology of PD.

In PD gut metagenomes, we detected an enrichment of type III, IV, and VI secretion systems, which are a hallmark of pathogenic bacteria. Our finding aligns well with previous studies reporting an enrichment of potentially pathogenic bacteria in the PD gut microbiome[15,29], which we partially replicated here. Secretion systems are used by pathogenic bacteria to, amongst others, modulate the host immune response during infection and can cause an activation of inflammatory response and an increase in gut permeability[52,53]. As a first line of defence in the innate immune response against infections, the host can produce CAMP, which are broad-spectrum antimicrobials also involved in modulating inflammatory responses[35]. Hence, the enrichment of systems used by bacteria to resist CAMP, which we detected here, suggests elevated host defence levels against potential infective agents in the gut of PD patients. Infective agents can lead to an increase in gut inflammation and permeability, which are both commonly observed in PD patients[49]. This deterioration in gut health might then contribute to the translocation of proinflammatory signals and cells to the CNS[54,55]. Finding these functions enriched in the PD gut metagenomes across study populations is highly relevant as it suggests new hypotheses on how the gut microbiome might contribute to the deterioration of gut health and favour the spread of pathogenic processes along the gut-brain axis. Recently, the connection between gut microbiota, gut health and CNS has emerged as an important aspect affecting neurodegeneration and ageing. For example, faecal microbiota transplantation between aged and young mice showed that the aged donor microbiota increases gut permeability, and systemic inflammation and accelerates age-associated CNS inflammation in young mice[56]. Interpreting our results in light of these recent experimental findings, we hypothesise that the gut microbiome of PD patients has an increased pathogenicity potential, which could trigger a pro-inflammatory response and compromise the integrity of the gut epithelial barrier. Compromised gut health and integrity can then facilitate epithelial translocation of toxic compounds (including chemicals, see below) and bacterial proteins, such as curli fibres allowing them to more easily reach the CNS. There they could stimulate αSyn aggregation, Lewy's body formation, neuronal toxicity, and neuroinflammation. Further experimental evidence is required to verify whether and to what extent these processes might impact PD development.

Strikingly, the extensive functional metagenomic analyses we performed here revealed many microbial pathways and enzymes involved in xenobiotics degradation to be enriched in PD metagenomes. Although some enriched genes, such as *xylC, todC1, todB* (KOs: K00141, K03268, K18089) are found exclusively in KEGG xenobiotic metabolism, they can be involved in the degradation of multiple molecules (Toluene, Nitrotoluene, Xylene, ɣ-Hexachlorocyclohexane, TCE). Hence, it is not possible to pinpoint the specific xenobiotic types that might have contributed to selecting these signatures. However, the enrichment of pathways involved in xenobiotic degradation suggests that the PD microbiome has been exposed to and has adapted to these chemicals. Although we cannot exclude that the enrichment of these pathways is a microbiome adaptation to the medications taken by PD patients, our findings align well with current epidemiological data indicating that exposure to such environmental xenobiotics is an important risk factor for developing PD[4,39–42]. There are several conceivable ways in which the observed alterations in gut microbial xenometabolism may be an adaptation to and/or actively modulate environmental exposures. On the one hand, the composition of the gut microbiome might be directly altered as a consequence of exposure to these chemicals[57,58]. In agreement with this first hypothesis, recent experimental data showed that rats exposed to TCE showed signs of PD pathology[41] and a concomitant gut microbiome enriched in *Bifidobacterium* and a depleted in *Blautia*[42], similar to the microbiome changes we observed here in human PD patients. On the other hand, it is an intriguing question if or to what extent gut microbial metabolization alters the toxic effects on dopaminergic neurons and the neuroinflammation that some of these chemicals induce[41,42]. Are gut bacteria producing more or less toxic metabolites during the catabolism of these xenobiotics? Besides a potential detoxification ability of

the gut microbiome, it is not unlikely that, instead, some microbial catabolites may have increased toxicity, as has been reported for some industrial chemicals and food dyes[57,58]. Since the gut microbiome is characterised by high inter-individual variability, it might represent a person-specific risk modulator of xenobiotic exposures. This implies that some people exposed to xenobiotics might have a higher likelihood of developing PD due to specific gut microbial metabolic capabilities resulting in increased neurotoxicity, whereas others may benefit from gut microbial detoxification of environmental chemicals. Further work integrating exposure and microbiome data with experimental work on microbial xenometabolism is warranted to shed light on the complex interactions between these two important factors. In summary, our data provide the most comprehensive overview to date about the taxonomic and functional alterations of the gut microbiome in PD patients and provide future reference for its use as a diagnostic tool.

## Methods

### Selected datasets

We collected 16S rRNA gene amplicon (16S) and shotgun metagenomics (SMG) datasets related to case-control studies that compared the composition of the gut microbiome between PD and control groups. We include all studies irrespective of the inclusion/exclusion criteria used, the typology and severity of PD, and the country of origin. We identified a total of 52 studies from which we excluded all studies that profiled < 30 samples, did not make raw data available, or for which it was not possible to assign the samples to patients or controls due to the lack of basic metadata. We could match the study of Hopfner et al. [59] with ENA's Bioproject PRJEB14928 and included this study in our analyses as well. In total, we collected 22 datasets, of which 16 and 6 studies profiled the gut microbiome using 16S and SMG sequencing, respectively. To perform a cross-disease comparison of the ML models built for the 16S data, we additionally included datasets related to multiple sclerosis[60–64] and Alzheimer's disease[65–69]. We performed this test using only 16S data due to the limited availability of SMG data for other neurodegenerative diseases.

### Profiling of 16S amplicon and shotgun metagenomic data

All 16S data were analysed using the DADA2 algorithm[70], yielding amplicon sequence variants (ASVs). When present, primers were removed either using cutadapt[71] v_3.4 or within the DADA2 workflow. Trimming parameters were adjusted for each dataset to meet the different quality of the data. Samples sequenced on different runs were profiled independently to allow a run-specific estimation of the sequencing error rates. The data from Wallen et al. [29] were sequenced using two different approaches, one using 150 bp and the other 250 bp reads length. Hence, they were split (Wallen151 and Wallen251) and analysed independently (which resulted in a total of 17 16S datasets). Taxonomy was assigned using Naive Bayes classifiers and the GTDB v_207[72] database. Finally, data were combined at the genus level while samples with < 2000 reads were discarded.

Taxonomy profiling of the SMG data was performed using mOTUs v_3.0[73]. For simplicity, we here refer to mOTUs as species, unless otherwise specified. The data from Boktor et al. [16] contained two independent datasets, which we analysed separately (Boktor_1, Boktor_2; which resulted in a total of 7 SMG datasets). The mOTUs taxonomy was then matched with the GTDB v_207 taxonomy using previously published mapping files (https://github.com/motu-tool/mOTUs/wiki/GTDB-taxonomy-for-the-mOTUs). Data were transformed into relative abundances and "unassigned" read counts were removed.

Functional profiling of the shotgun metagenomic data was performed using gffquant v_2.10 (https://github.com/cschu/gff_quantifier) in combination with a reduced version of the GMGC human gut nr95 catalogue[74] obtained by removing genes that only occurred in less than 0.5% of samples used for building the original human gut catalogue. This reduced the catalogue to 13,788,251 non-redundant genes. Prior to functional profiling, raw reads were cleaned using bbduk[75] v_38.93 as follows: (1) low-quality trimming on either side (qtrim = rl, trimq = 3), (2) discarding of low quality reads (maq = 25), (3) adaptor removal (ktrim = r, k = 23, mink = 11, hdist = 1, tpe = true, tbo = true against the included bbduk adaptor library) and (4) length filtering (ml = 45). The cleaned reads then were screened for host contamination using kraken2[76] v_2.1.2 against the human hg38 reference genome with ribosomal sequences masked (Silva[77] v_138). The remaining reads were finally mapped to the reduced human gut gene catalogue using BWA-MEM[78] v_0.7.17 with default parameters and name-sorted by samtools[79] v_1.14 *collate*. The resulting alignments were filtered to > 45 bp alignment length and > 97% sequence identity. Reads aligning to multiple genes contributed fractional counts towards each matching gene. Alignment counts for a gene were normalised by the gene's length, then scaled according to the strategy employed by NGLess (https://ngless.embl.de/Functions.html#count) and propagated to the functional features with which the gene is annotated. The final counts were normalised by dividing against the sum of all mapped reads passing our filtration criteria to obtain relative abundances. For KEGG KOs, we retained only KOs of prokaryotic origin according to KOFAMKoala[80] prokaryotic HMMs. We additionally filtered both KEGG pathways and modules by retaining those consisting of at least 50% and 60% prokaryotic KOs, respectively.

Gut microbial modules (GMMs)[27] and gut-brain modules (GBMs)[28] were inferred based on KOs via the R package omixerRpm[81] v_0.3.3 using default parameters and a pathway coverage (*minimum.coverage*) of 0.5. We then used the KEGG mapper[82] portal to map the differentially abundant KOs onto the KEGG pathway maps and verify in which xenobiotic metabolisms they are involved. Finally, we used the protein sequence of the TyrDC enzyme encoded by *Enterococcus faecium* (NCBI ID: QAV53956) to verify whether this enzyme is encoded in the genomes of the lactic-acid producing bacteria enriched in PD. The protein sequence was used to query the NCBI database through *blastp*[83].

### Statistical analyses

All data analyses were performed in R[84] v_4.2. For both taxonomic and functional profiles relative abundances were used for further analyses. First, for both 16S and SMG data ordinations were built based on Bray-Curtis dissimilarities using the phyloseq[85] v_1.40 and vegan[86] v_2.6.4 R packages. Specifically, ordinations were built using distance-based redundancy analysis (dbRDA) implemented in the *capscale* function within phyloseq as previously described[12], with and without conditioning the data by study. The significance of the clustering (for study of origin, disease condition, country, continent, and Western vs Eastern origin) was tested on the Bray-Curtis dissimilarities using permutational multivariate analysis of variance (PERMANOVA, *adonis2* function; with 2000 permutations). PERMANOVA for the disease status was performed by restricting the permutation within datasets. Differences in Bray-Curtis dissimilarities within studies and between studies were tested using a two-sample *t* test (two-sided).

All differential abundance analyses were conducted on filtered features, retaining only those for which a minimum prevalence of 5% was observed, with the exception of the analyses done for the GMMs and GBMs for which data were not filtered. This corresponded to 202 genera (obtained from 16S data), 1808 mOTUs, 7632 KO, 581 KEGG modules, 144 KEGG pathways, 103 GMM, and 49 GBM. Agresti generalised odds ratios (*genodds*[87] v_1.1.2 R package) were used to estimate effect sizes and standard errors in each independent dataset. This statistic, analogous to the U statistic underlying the Mann–Whitney test, is based on ranks and does not make strong assumptions about data distributions. It calculates the odds of the second group having a higher value of the outcome (taxa abundances in our case) than the

first group if a pair of observations are randomly selected from a dataset. We used the default settings for tie splitting to obtain odds ratios that are equivalent to the Wilcoxon-Mann-Whitney odds ratios. Estimates were then pooled using random effect meta-analysis (meta[88] v_6.2.1 R package), with *p-values* adjusted using the Benjamini−Hochberg method (False-Discovery Rate, FDR). Adjusted *p-values* are referred to as *q-values* here. For the functional data, we additionally performed a gene set enrichment analysis using the generic *enricher* function in the R package clusterProfiler[89] v_4.4.4. This was used to perform independent hypergeometric tests on the subset of KOs enriched either in PD or in CTRL with the aim of estimating which KEGG pathways were significantly enriched in differentially abundant KOs. Background genes (or universe) were defined as all KO within the KEGG pathways that were represented in our dataset. Enrichment tests were run using *minGSSize = 5*, *maxGSSize = 500*, *p-values* were adjusted using FDR, and alpha was set to 5%. Finally, Pearson correlations were calculated to assess consistency between Ridge regression relative weights and the generalised odds ratios.

Due to the sparsity of available metadata, we used a subset of datasets to perform a sensitivity analysis and identify microbiome features that might potentially be confounded by donor covariates such as age, sex, or medication usage (SMG: 50% of the datasets provided sex and age information, 17% medication; 16S: 56% sex and age). Although age and sex are risk factors for PD and thus intrinsically associated with the disease[1,4], we included them in this analysis to account for sampling biases. These analyses were performed for all microbiome features we detected associated with PD in our meta-analyses. To test the effect of medication usage we applied two independent strategies. First, we selected all metadata related to medication usage available from Wallen et al.[15]. We then retained only medications used in at least 20% of the participants (11 medications in total) and used them to perform a variable selection using the *regsubsets* function in the leaps[90] v_3.1 R package. This was done for the regression analysis modelling the abundance of the features as a function of medications and disease status, allowing models with a maximum of 12 variables (including all medications and the disease status). We then selected the variables defining the regressions with the minimum Mallow's Cp value and used them to build the final linear models (lm_covariates; *feature ~ medications + PD*; where the term *medications* can include up to the 11 medications we considered). Additional baseline linear models were built for the same dataset including only the disease status (lm_pd; *features ~ PD*). FDR-corrected *p-values* were then compared between model types (lm_pd vs lm_covariates). All features with a significant association with PD (*q-values* in the lm_pd models < 0.05) which were affected by the correction for medication intake (PD *q-value* in the lm_covariates ≥ 0.05) were considered as potentially confounded. Second, we selected all metadata on PD medications available for the study of Boktor et al.[16,91] and build linear mixed models for each medication (*feature ~ medication + (1 | cohort)*). After correcting *p-values* using FDR, we selected as potentially confounded all those features that had a statistically significant association with at least one PD medication. While these analyses suggested some features to be confounded (Supplementary Datas 6–9), we need to note that in particular for PD medication this analysis may not be well-powered to detect all confounding effects. For a more thorough confounder analysis, more complete data on the medication of PD patients is required. Finally, we tested the confounding effect of sex and age by comparing the significance of the association between microbiome features and PD before (baseline models; *feature ~ PD + (1 | cohort)*) and after accounting for covariates (*feature ~ sex + age + PD + (1 | cohort)*). This analysis was performed for both SMG[15,16,18] and 16S[21,24,25,29,92–96] datasets with available metadata. Metadata from the study of Bedarf et al. [18] were obtained from the repository related to the study of Boktor et al. [91]. After correcting *p-values* using FDR, we selected as potentially confounded all those features having a

significant association with PD in the baseline models (*q-value* < 0.05) which became statistically non-significant after accounting for covariates (*q-value* ≥ 0.05). All the above analyses were conducted on log-transformed relative abundances, and linear models were built using either the *lm* from the stats[84] v_4.2.3 R package or the *lme* function from the nlme[97,98] v_3.1.162 R package.

### Machine learning approaches

Machine learning models were built using the SIAMCAT v_2.0 and v_2.10 toolbox[23]. Model accuracy was assessed using a 10-times repeated 10-fold cross-validation ($10 \times 10$ CV) unless otherwise stated. We built models using all machine-learning algorithms provided through SIAMCAT (Ridge regression, Elastic Net, LASSO, Random Forest, as well as Ridge regression and LASSO as implemented in LibLinear[99]). SIAMCAT workflows included an internal hyperparameter tuning step (via a cross-validation approach that is applied to the respective training data and nested into the out cross-validation). We assessed model performances on data normalised using either log transformation (log.std) or centred log ratios (clr). The performance of all ML models was quantified by the area under the receiver operating characteristics curve (AUC). For repeated cross-validation, sample classification probabilities were averaged across repeated runs and used to estimate a final AUC. The effect of feature filtration on model accuracies was assessed by building models using all the above-indicated algorithms on datasets filtered to retain only the most commonly detected and prevalent taxa. Specifically, we used datasets filtered by discarding all taxa detected in less than 5%, 10%, 20%, and 30% of the samples in 10 and 2 datasets for the 16S and SMG data, respectively. For all ML algorithms tested, study-to-study validation (cross-study validation; CSV) was performed by testing the models built on each dataset on every other dataset. Leave-one-study-out (LOSO) validation was performed by combining all but one dataset at a time. The combined data were then used to train Ridge regression models in $10 \times 10$ CV following the strategy implemented in SIAMCAT. The left-out study was used to test model performances. From the 10 repetitions of within-study CV and the resulting 100 models of each LOSO run, averages and standard deviations of AUCs were computed and displayed in Supplementary Fig. 6. Differences in AUCs between validation strategies and ML approaches were tested using a two-sample Welch *t* test (two-sided), the correlation between AUCs and training set sizes were assessed using Pearson correlations. Finally, we identified a subset of species (mOTUs) that could robustly discriminate PD from controls. To do this, we conducted a feature selection based on a differential abundance analysis independently for each of the 7 training sets used for the LOSO validation. Within each training set, we identified differential features using a two-sided Wilcoxon-Mann-Whitney (WMW) test with blocking by study (R package coin[100] v_1.4.2). Within each of the 7 training sets, we then selected the 20 features with the highest absolute effect size (test statistic from the WMW test) and significant difference between PD and controls (FDR adjusted *p-values* < 0.05), and used them to perform new LOSO validations as explained above.

To investigate the effect of dataset pooling on LOSO validation accuracy, we performed 7 independent combinatorial analyses of training set composition, one for each SMG study used as a hold-out test set. In this approach, we trained a single Ridge regression model on every possible combination of pooled datasets, progressively increasing the number of combined studies from two to six. This resulted in a total of 57 different training sets and, hence, in 57 independent Ridge regression models tested on the same hold out set. The association between LOSO AUCs and a number of samples was then assessed using linear mixed-effect models with the test set as a random intercept. Marginal $R^2$ (corresponding to the proportion of variance explained by the fixed effect, number of samples in this case), as well as the intraclass correlation coefficient (which can be interpreted as the

proportion of variance explained by the mixed effect alone), were extracted using the R package performance[101] v_0.11. For this analysis, the number of samples used for training was first log-transformed and then scaled. Finally, to directly compare model performances derived from 16S and SMG data, we used the data from Jo et al. [19] where both data types had been generated from the same samples. We built Ridge regression classifiers using 10 × 10 CV with identical sample splits between testing and training sets for each data type. Model performances were then directly compared using AUCs, as previously specified.

ML models for the functional profiles were built by applying the prevalence filtration described above at the 5% threshold. For the GMMs and the GBMs, no filtration was applied. For model building from KOs, we initially run Ridge regression models using a 5 × 5 CV to identify the best subset of features to use for training. Using a nested supervised feature selection based on the Wilcoxon test within SIAMCAT (as described above), we built models allowing from 500 to 4000 features (in steps of 500). We then selected the number of features that resulted in the highest median AUC across datasets (AUC = 75.3, 2500 features), and used it to build and evaluate final 10 × 10 CV models. The same number of features was also used to perform a LOSO validation with a nested supervised feature selection as described above. Differences in AUCs across SMG profiles was assessed through linear models using the nlme[97,98] v_3.1.162 R package with the training-test set combination as a random intercept. Contrasts were extracted using the emmeans[102] v_1.8.5 R package and *p-values* were adjusted using FDR. We additionally investigated the effect of differences in age and sex distribution between cases and controls, as well as geographical study origin (Western vs Eastern) on the AUCs of within-study CV and CSV accuracies obtained for the taxonomic profiles of both 16S and SMG data. As a summary statistic for age, we computed the ratios of the average ages of PD and control donors within studies. For sex instead, we first calculated the ratios of the number of female (F) and male (M) donors in PD and controls within studies. We then used these F/M values to compute ratios between the control and PD samples. All AUCs for the CSV were then split based on the (Western vs Eastern) origin of the training and test set (e.g., W_W when both training and test set came from Western populations). Similarly, AUCs of the within-study CVs were divided based on the W/E origin of the studies. The associations between these population features and AUCs were then assessed using linear models (e.g., *AUC ~ age.ratio*) and $R^2$ were extracted. Finally, for all Ridge regression models derived from single datasets, we extracted the model's weights (Ridge regression coefficients) and divided them by the absolute sum of all feature coefficients to calculate relative weights. Relative weights for each feature were then summarised in the figures by average and standard deviation calculated across datasets. To visualise model similarity across studies, the coefficients were further used to create a non-metric multidimensional scaling (NMDS), based on Canberra distances. The effect of the continent of study origin on the clustering of the models (one for each study) was tested using PERMANOVA.

To test the effect of sequencing depth on the accuracy of the 16S-based models, we additionally rarefied the data to a depth of 2000 reads using the rtk[103] v_0.2.6.1 R package. ML models were then built and evaluated through the same workflow as described above (both CV and CSV) and compared to models built on non-rarefied data using a paired *t* test. For all the t-tests performed in this study, Cohen's D effect sizes and their 95% confidence intervals, were estimated using the *cohens_d* function in the rstatix[104] v_0.7.2 R package. Moreover, the removal of study heterogeneity from the 16S data was performed using the function *adjust_batch* with default parameters in the MMUPHin[105] v_1.10.3 R package as well as the function *ba* in the bapred[106] v_1.1. R package using the methods: *meancer*, which centres the variables within batches (datasets in our case) to have zero means (to remove negative values we added to the data the negation of the lowest

corrected abundances); *ratiog*, which divides the variables by the batch-specific geometric mean of the corresponding variable; *ratioa*, which divides variable values by the batch-specific arithmetic mean of the corresponding variable. A batch is here considered equivalent to a study. For each batch correction method, we used the study-specific models to perform independent CSV. Differences in AUCs across ML methods was assessed through linear models using the nlme[97,98] v_3.1.162 R package with the training-test set combination as random intercept. Contrasts were extracted using the emmeans[102] v_1.8.5 R package and *p-values* were adjusted using FDR. Correlations between rarefied and not rarefied data were tested using the *cor.test* R function (Pearson correlation). Finally, to perform a cross-disease validation, we tested all study-specific and LOSO 16S Ridge regression models on additional 16S datasets obtained for other neurological diseases. False positive rates (FPR), representing the proportion of samples in the test dataset predicted wrongly as PD were then extracted as previously described by Wirbel et al. [13]. For the LOSO models, the FPRs were extracted from the held-out test set. We restricted this analysis to 16S datasets due to the scarcity of SMG data for other neurological diseases.

## Reporting summary

Further information on research design is available in the Nature Portfolio Reporting Summary linked to this article.

## Data availability

All data used in the article are either publicly available or have been directly obtained from the authors of the original publications, as summarised in Table 1. PRJNA601994; CRA001938; PRJNA494620; PRJNA381395; PRJNA391524; PRJNA268515; PRJNA510730; PRJEB27564; PRJEB30615; PRJEB14928; DRA009229; PRJNA808166; Kenna et al. [107] https://doi.org/10.6084/m9.figshare.14345513.v1[108]; PRJNA742875; PRJEB17784; PRJNA433459; PRJNA588035; PRJNA743718; PRJNA834801; ERP138197 [https://www.ebi.ac.uk/ena/browser/view/PRJEB53401]; ERP138199 [https://www.ebi.ac.uk/ena/browser/view/PRJEB53403]; PRJNA489760; PRJNA633959; PRJNA321051; PRJNA450340; PRJEB34168; PRJNA721421; PRJEB99111; PRJNA554111; PRJNA734525; PRJEB51982; metadata Boktor et al. [16] [https://zenodo.org/records/7183678][91]; metadata Wallen et al. [15] [https://zenodo.org/records/7246185]. The gut microbiome taxonomic and functional profiles generated in this study are available on Zenodo https://doi.org/10.5281/zenodo.14261087.

## Code availability

The R code used in this manuscript is publicly available on GitHub[109] at https://github.com/StfnRomano/PD_ML_meta.

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

## Acknowledgements

The authors are indebted to all colleagues who made microbiome data and metadata available for re-analysis. They are moreover grateful to Michael Zimmermann and members of the Zeller group for inspiring discussions. In addition, we thank Y. P. Yuan, J. Pečar and the EMBL IT Services team for support with high-performance computing. This research was supported by the Biotechnology and Biological Sciences Research Council (BBSRC) through its Institute Strategic Programme Gut Microbes and Health BB/R012490/1 and its constituent project BBS/E/F/000PR10356. S.R. was partially funded by an EMBO Scientific Exchange Grant (grant no. 9093). G.Z. is supported by EMBL, LUMC, the Federal Ministry of Education and Research (BMBF grant no. 031L0181A), the Deutsche Forschungsgemeinschaft (DFG, German Research Foundation no. 395357507 – SFB 1371) and an LUMC Fellowship. The funding bodies had no role in the study design, execution of the analyses, and data interpretation. Q.D. was supported by a Health + Life Science Alliance Heidelberg Mannheim through state funds approved by the State Parliament of Baden-Württemberg and an EMBO postdoctoral fellowship (EMBO ALTF 1030-2022).

## Author contributions

S.R. conceived the project, conducted bioinformatic and statistical analyses, acquired funding, and draughted the manuscript. J.W. supported statistical data analyses and contributed to the manuscript. R.A. supported data analysis, visualisation, and interpretation and contributed to the manuscript. Q.D. performed functional profiling and contributed to the manuscript. C.S. developed the software to perform functional profiling. A.N. provided financial support and helped with data interpretation. G.Z. supervised the work, advised on data analysis, visualisation and interpretation, contributed to the manuscript, and acquired funding. All authors read and approved the final version of the manuscript.

## Competing interests

The authors declare no competing interests.
