## [Transparent Peer Review file · Nature Communications]

Machine learning-based meta-analysis reveals gut microbiome alterations associated with Parkinson's disease.

Corresponding Author: Dr Stefano Romano

Version 0:

Reviewer comments:

Reviewer #1

(Remarks to the Author)

The study by Romano et al conducted a machine learning meta-analysis of gut microbiome studies of Parkinson's disease (PD) based on large-scale 16S amplicon and shotgun metagenomics sequencing (SMG) datasets from multiple populations. The study revealed that the machine learning for most studies could not generalize well across other studies, while models pooling multiple datasets improved their general applicability. In general, the study offers a comprehensive overview about the gut microbiome of PD and strengthens the findings by individual studies. Moreover, the study identified several taxa and microbial pathways associated with PD, which may contribute to the deterioration of gut health and the translocation of pathogenic molecules along the gut-brain axis. However, the study did not characterize a significant finding or methodological novelty, and did not well document how these findings help to enhance our understanding of the mechanisms, subtyping, diagnosing or treatment of the disease. I have the following major concerns.

First, there're problems in organizing of the paper. 1) Abstract. The abstract provides no analytic output but lists all conclusive comments. Crucial outcomes of the analyses must be presented here, including results concerning the comparison of prediction accuracies of the ML models for within-study CV, CSV, LOSO studies, the most enriched taxa in PD, etc. Moreover, the conclusion seems not clear. 2) Results. The Results part was set to display findings of the study. Adding a large amount of discussions plus references here is not a regular way. In particular, the large amount of comments added in the last part of the results was too redundant and need to be greatly simplified. 3) Discussion. This part should focus more on the novelty of the ML methods, novel disclosures that contribute to the mechanisms, diagnosis or therapies to the disease, and the explanations of the improved prediction accuracies of the pooling multiple analyses versus for single studies.

Second, several crucial scientific issues need to be addressed. 1) The considerable heterogeneity of PD involves much more factors than analyzed in the study, which at least include age, sex, BMI, life styles (e.g. smoking, drinking, eating), environmental exposures, comorbidities (e.g. diabetes, hypertension, hyperuricacidemia), disease duration, stage, subtypes, and medications. The in-depth analyses need to be performed to provide better interpretations for the low study-to-study portability, and possibly, identified factors that most severely affect the gut microbial profiles in PD. 2) The Westerners and Easterners have huge epidemiological differences, including dwelling conditions and life styles. The analyses did not investigate the common and different microbial profiles concerning the two distinct populations. 3) Although several microbiome signatures have been proposed to be commonly enriched or depleted in PD, the study did not propose a PD-specific ML models that helps to the differentiation of this disease from other neurodegenerative diseases. 4) The abundance of lactic-acid producing bacteria (e.g. Lactobacillus sensu-lato, Bifidobacterium, and Ruthenibacterium) may not be attributed only to the use of L-dopa, more discussions are required to interpret these findings.

(Remarks on code availability)

Reviewer #2

(Remarks to the Author)

Romano et al. present a comprehensive study on the application of machine learning (ML) techniques to diagnose Parkinson's disease (PD) using gut microbiome data. It explores the effectiveness of various ML models, comparing the

diagnostic accuracy of taxonomic and functional profiles of the microbiome. The study also investigates the impact of data preprocessing methods, such as sequencing depth and normalization, on model performance. Additionally, it addresses the challenge of model generalizability across different datasets and the specificity of microbiome-based diagnostics in distinguishing PD from other neurodegenerative diseases. Finally, it identifies the gut microbial taxa and their functions that are strongly associated with PD. This review paper is interesting to me. I am quite open to looking at a revised version if the authors could address some major and minor issues in a satisfactory fashion, which I describe in more detail below.

Major issues:

1. Regarding the claim "The study of origin instead explains a considerably higher proportion of the data, 19.9% and 7.7% for the 16S and SMG data, respectively", are there any figures to visually demonstrate this? For example, the authors can generate a similar distance-based redundancy analysis while all data points are colored by the study of origin to show how the data from different study origins separate.
2. The training of machine learning (ML) methods, such as random forests, is not clear to me. How do the authors select the training data versus the test data? Did they use the random data splits? Is the presented performance obtained by measuring the predictive power of the test data? Additionally, I don't know how the hyperparameters are determined exactly. Is the cross-validation of the training data used to select the hyperparameters? I think a better clarification of the training, validation, and test has to be included in the "Comparison of machine learning approaches" and the method sections.
3. I worry that the comparison between the average AUC (area under the receiver operating characteristics curve of predicting PD or control based on the 16S taxonomic profile and that based on the shotgun taxonomic profile in Fig. 2 might be heavily based on the difference in sample size across datasets. Presumably, a large dataset would have more samples to train the ML algorithms and thus lead to better predictive performance on the test data. Therefore, the comparison between two cases with different sample sizes can cause erroneous results. I would suggest they only select samples with paired 16S and shotgun metagenomic data and then use the matched samples with paired data to investigate if the predictive performance using the 16S taxonomic profile is different from that using the shotgun taxonomic profile.
4. Regarding the comparison of AUC for the within-study cross-validation (CV) with the AUC for the leave-one-study-out validation (LOSO), I wonder whether the test data for both cases are the same. Specifically, when the authors measure the CV predictive performance on the same dataset, in principle, they use a fraction of the dataset for the test. Is it the same as the test set used in the LOSO? They need to be the same so that the performance benchmarking is meaningful.
5. It is great to see the comparison of AUC for CV with that for the LOSO. However, I think the usefulness of pooling data across studies needs to be investigated from a different perspective: is pooling more independent datasets across studies better than only pooling a few? I believe it is important to change the number of independent datasets combined as the training set and study how the AUC for CV changes as the number of independent datasets used in the pooling varies. In this way, it helps explain how many datasets across studies should be pooled to enhance the predictive performance on a separate dataset.

Minor issues:

1. I am confused about why the p-values for all cases in Fig. 1 are 0.001. Should it be < 0.001 ? It would be better to show the raw p-values.
2. Some figure labels are wrongly referred to in the text. For example, in the section "Cross-study portability of the ML models," when the authors describe the results of 16S, they should refer to Fig. S3 instead of Fig. S4. Similarly, when they describe the results of the shotgun taxonomic profile, they should refer to Fig. S4 instead of Fig. S5. Please check the figure labels throughout the article.
3. On page 8, "all Ridge models built on the 16 and SMG data" should be "all Ridge models built on the 16S and SMG data".
4. It seems to me that Fig. 5 looks vertically compressed. Please find a way to fix this issue.

(Remarks on code availability)

The code itself looks fine. But it only contains the code used for data analysis. No data processing is included. Thus, it is hard for a scientist to leverage the code to deal with a similar scientific analysis from raw data. I would recommend including an example in the README file to elaborate how to perform a similar data analysis based on raw data.

Reviewer #3

(Remarks to the Author)

In this article the authors, present an extensive meta-analysis of 16S and SMG data from numerous studies using a ML approach to explore the microbiome signature implicated in Parkinson's disease. They provide several hypotheses while analyzing the signatures and overall models's performance. The article is easily readable, although some of the figures could be improved.

- Figure 1, For better lisibility it may be useful to reduce size of the points and eventyally remove the outlines from the boxplots or lower their size. The first two dimensions of the MDS explain little and this could be emphasized in the text. Another comparative approach could be the intra-study vs inter-study sample differences... Have the authors evaluated the impact of geography alone rather than the study ? Have the authors explored the specific effect of preprocessing protocols, which is known to be a strong determinant of inter-study variability ? They mention that they investigated batch effects in the 16S data. How were these batches identified?

- Concerning Figure 2 the authors write that: "For both 16S and SMG data the LOSO model performances were significantly higher than those obtained through CSV". Are they refering to the gray points ? This annotation can be confusing with the CV (blue) and could benefit from being added in the Figure's 2 legend. What is the CV schematic? 10-fold ? If there is more than one fold, it would also be interesting to illustrate the variability using a violin plot as well? (This is done in Figure 4)

- Figure 3 is difficult to read. Looking at the distribution of the violinplots I'm not sure that averaging results is the best way to

present these results. There are clearly some models (datasets), which work much better than others. Are these 16S or SMG data/models? Until now the data results on data types were presented separately (after going through the text it looks like 16S) - this information could be specified also in the legend. The gray lines in the orange violinplots are barely visible. It could be interesting to add the AD and MS classification results in two other panels in Figure 2 and also the PD datasets in Figure 3 as a baseline. Moreover, these diseases and data could be introduced before in the manuscript.

- The authors write: "In our meta-analysis, we found that models built on taxonomic profiles perform in general better than those built on functional profiles (Fig 4)". However from this figure the variability of the performance is large and it is not clear how statistically significant this is. Can the authors provide more statistics backing up this claim?

- I understand that the feature spaces are different between WGS and 16S thus making comparison of the models difficult. But have the authors tried to bring these two spaces together?

- Figure S2, it is not clear whether the results illustrated here concern only for the two studies mentioned in the legend or all the studies? Idem for Figure S1.

- Figure S3 is difficult to see patterns, maybe some kind of hierarchical clustering could help improve visibility.

- Figure S6: please specify the metric used. What about the SMG dataset, what is the impact of rarefaction there? How does the rarefaction fluctuate when changing the seed?

- Figure S8: not sure that the model weights in the ridge algorithm is the best descriptor to compare datasets. A more consensual descriptor in terms of predictability could be the feature importance for instance. In this case the authors can not conclude much, since there are issues with the modeling itself. Also ellipses could improve/facilitate the interpretation.

- The co-variate analysis was restricted to age, sex, the authors also mentioned medication usage. Can they report for which percentage of the cohorts was this available and at what resolution? The lack of data can explain the low rate of variables affected by these confounding.

- Figure 6b, please specify what the black rectangles indicate.

- The interpretation of the results on the functional spaces, although interesting, is only based on data that indicate the functional potential, which is also difficult to estimate as the different modules and pathways may be complete or not are used to compute the relative abundance. It is important not to oversell here and prevent speculation.

- Can the authors comment on the soundness of the covariate analyses feature ~ sex + age + PD + (1|cohort), knowing the distribution of microbiome features?

- Concerning data availability, the authors state: "All data used in the article are either publicly available or have been directly obtained from the authors of the original publications, as specified in Table 1.". However, the authors have put considerable effort in analyzing these public dataset and computing the different taxonomic and functional profiles. It would be quite useful to the scientific community that these processed data (taxonomic, functional, metadata, and code) are easily accessible to the community, the same was as the original data were accessible to them.

(Remarks on code availability)

I briefly explore the code without testing it

- Concerning data availability, the authors state: "All data used in the article are either publicly available or have been directly obtained from the authors of the original publications, as specified in Table 1.". However, the authors have put considerable effort in analyzing these public dataset and computing the different taxonomic and functional profiles. It would be quite useful to the scientific community that these processed data (taxonomic, functional, metadata, and code) are easily accessible to the community, the same was as the original data were accessible to them.

Version 1:

Reviewer comments:

Reviewer #1

(Remarks to the Author)

The authors have addressed all my comments.

(Remarks on code availability)

Reviewer #2

(Remarks to the Author)

I appreciate that the authors have responded satisfactorily to most of my comments. Yet, I still have concerns over their response to my previous major point 3. The presented Fig. S7 needs to be more convincing because different sample sizes across datasets cannot be used to support their argument that the training set size does not strongly influence the AUCs. Instead, I bet if they use the same dataset and vary the training sample size, the training sample size will strongly influence the AUC.

I understand that it is hard to have a matched comparison between 16S and shotgun metagenomics datasets. However, as the authors mentioned in their response, the Jo et al. dataset includes 16S and metagenomics data. I believe the authors can (1) find the shared samples with both 16S and metagenomics data, (2) leave the same samples out for the test data, and (3) then perform a one-to-one comparison between 16S and metagenomics data using the same training sample size. It is better to show this comparison directly as a figure to demonstrate their claim that using the metagenomic data can generate

better predictive performance than using the 16S data.

(Remarks on code availability)

I think the revised README.md provides more comprehensive information. I do not have further comments.

Reviewer #3

(Remarks to the Author)

I appreciate the authors' work in answering all my comments and taking into consideration most of the suggestions.

(Remarks on code availability)

REVIEWER COMMENTS

Reviewer #1 (Remarks to the Author):

The study by Romano et al conducted a machine learning meta-analysis of gut microbiome studies of Parkinson's disease (PD) based on large-scale 16S amplicon and shotgun metagenomics sequencing (SMG) datasets from multiple populations. The study revealed that the machine learning for most studies could not generalize well across other studies, while models pooling multiple datasets improved their general applicability. In general, the study offers a comprehensive overview about the gut microbiome of PD and strengthens the findings by individual studies. Moreover, the study identified several taxa and microbial pathways associated with PD, which may contribute to the deterioration of gut health and the translocation of pathogenic molecules along the gut-brain axis.

However, the study did not characterize a significant finding or methodological novelty, and did not well document how these findings help to enhance our understanding of the mechanisms, subtyping, diagnosing or treatment of the disease.

I have the following major concerns.

First, there're problems in organizing of the paper.

1) Abstract. The abstract provides no analytic output but lists all conclusive comments. Crucial outcomes of the analyses must be presented here, including results concerning the comparison of prediction accuracies of the ML models for within-study CV, CSV, LOSO studies, the most enriched taxa in PD, etc. Moreover, the conclusion seems not clear.

Response: *Thank you very much for the comment. We appreciate that a more precise summary of the key results may improve the Abstract. We had to balance the desire to add more information with the Journal's word limit.*

Action taken: *We have adjusted the Abstract following the reviewer's advice as much as possible within the word limit and have added the overall average AUCs for the CV, CSV, and LOSO approaches.*

2) Results. The Results part was set to display findings of the study. Adding a large amount of discussions plus references here is not a regular way. In particular, the large amount of comments added in the last part of the results was too redundant and need to be greatly simplified.

Response: *Thank you for this comment. Considering the amount of information and the diversity of findings (ranging from the ML results to taxa abundances and pathway analyses) we tried to explain and contextualise our main findings in the original version of our manuscript. We believe that this will help the reader to understand the bacterial taxa and functions we report and greatly increase readability (as compared to having explanatory bits in the Discussion only). We have nonetheless followed the reviewer's suggestion to emphasise presentation of the results and reduce text on their interpretation.*

Action taken: We have now reduced the interpretative parts in the Results and focused more on reporting the findings of our study. (Changes have been made in page 9 lines 17-23 and 26-32; page 10 lines 12-13 and 45-46, page 11 line 1).

3) Discussion. This part should focus more on the novelty of the ML methods, novel disclosures that contribute to the mechanisms, diagnosis or therapies to the disease, and the explanations of the improved prediction accuracies of the pooling multiple analyses versus for single studies.

Response: We appreciate the reviewer's wish to discuss the translational potential of our study, in particular how the microbiome can help us to better elucidate "mechanisms, diagnosis or therapies to the disease". However, while our work is certainly aimed at exploring the translational potential of the gut microbiome, we are convinced that it is also important to be careful in making conclusive statements about the underlying mechanisms or therapeutic applications – after all, our study is based on cross-sectional observational metagenomic data. The data we re-analysed came, in the majority of the cases, from case-control studies, which are not suitable for addressing mechanistic questions. However, since we agree with the reviewer on the importance of a mechanistic understanding, we have explored in great detail the metabolic functions encoded in the metagenomes of PD patients, which suggested new hypotheses on how the microbiome might contribute to the development of the disease. Specifically, we have provided insights into the potential role of virulence factors, and more importantly, xenobiotic metabolism. Nonetheless, to understand whether these bacterial functions have a real influence on the disease pathophysiology, additional experimental data will have to be collected. Similar considerations apply to the diagnosis aspect. As we have underlined in the discussion (original version of the manuscript: page 16, lines 22-32) data coming from PD patients at different stages and undergoing different medication regimes, have intrinsic biases which increase data heterogeneity and potentially influence the study-specific associations between microbiome and disease. Hence, we argued in the discussion that whether the microbiome is a strong marker for diagnosis of PD remains to be verified on data collected from early stage patients, possibly drug naive, or from high risk patients in prospective studies.

In any case, and directly addressing the reviewer's comment, we have further evaluated the possibility of general PD classification across studies using a small subset of species features, which we feel has further strengthened the basis for future efforts towards microbiome-based PD diagnosis.

Action taken: We have performed additional analyses in which we build new LOSO models using only a subset of features unbiasedly selected across training sets (Fig 3 and page 6 lines 32-46, page 7 lines 1-4). These new analyses show that even when only 20 species are used for model training, the prediction accuracies remain similarly high (as compared to full models), suggesting that these features may be generally useful for disease diagnosis.

Second, several crucial scientific issues need to be addressed.

1) The considerable heterogeneity of PD involves much more factors than analyzed in the study, which at least include age, sex, BMI, life styles (e.g. smoking, drinking, eating), environmental exposures, comorbidities (e.g. diabetes, hypertension, hyperuricacidemia), disease duration, stage, subtypes, and medications. The in-depth analyses need to be performed to provide better interpretations for the low study-to-study portability, and possibly, identified factors that most severely affect the gut microbial profiles in PD.

Response: *We agree with the reviewer that PD is a very heterogeneous disease and that, as in many microbiome studies, one must be careful about potential confounders (as stated in page 16 lines 22-32 of the original version of our manuscript). However, in our meta-analysis, as in many others, there are practical obstacles to implementing rigorous analyses accounting for confounding. Unfortunately, in many studies data on the factors mentioned by the reviewer are neither collected nor published and this cannot be rectified in our meta-analysis.*

Nevertheless, recognizing this issue, we have made a considerable effort to obtain crucial metadata for those studies where such data was published or made available by the authors on our request. In the original version of our manuscript we performed multiple analyses to verify whether the associations we detected between microbiome features and PD were potentially confounded by sex, age, and medication usage (see Fig S19; page 10 lines 24-27; page 12 lines 25-28; page 14 lines 22-24 and 38-39; page 15 lines 15-18 and 37-39; page 16 lines 32-38; page 21 lines 20-47; page 22 lines 1-10). As per the reviewer's request we have extended these analyses to an assessment of covariate effects on cross-study portability of the ML models.

Action taken: *We examined how differences in the sampled patient populations affect the portability of ML models. We included an additional figure (Fig. S8) in which we grouped the CV and CSV values based on three main features of the study population of origin: i) age distribution across cases and controls; ii) proportion of sexes among cases and controls; iii) origin of the study (Western vs Eastern). The associations between AUCs and these features were then tested using linear models/ANOVA. Our results suggest that none of these features strongly influenced model performance, indicating that rather other differences in e.g. medication usage, PD subtypes, stage and symptomatology might explain the lack of portability, as already discussed in the original version of the manuscript (page 16 lines 22-32). For example, when the study of Bedarf et al. was used either as training or as a test set, the accuracy for cross-study application of models was generally lower. This could likely be due to the fact that the patients in the study by Bedarf et al. are all early PD, L-DOPA naive, and all males. We have further stressed this point in page 11 lines 35-37 and page 12 lines 11-13 of the revised manuscript.*

2) The Westerners and Easterners have huge epidemiological differences, including dwelling conditions and life styles. The analyses did not investigate the common and different microbial profiles concerning the two distinct populations.

Response: *This is an interesting point. It is indeed well possible that geography-associated factors limit the generalisation of microbiome signatures and models, in particular between Westerners (W) and Easterners (E). We would like to point out, though, that it is difficult to directly investigate geography-associated differences in the gut microbiome in our meta-*

analysis setting, because technical differences between studies tend to be much more pronounced than geography-associated influences and are often aligned with these (see e.g. Fig. 2 in PMID: 33785070). This makes it difficult (often impossible) to clearly attribute the observed differences to geography-associated factors rather than less interesting technical study heterogeneity. Nevertheless, in our original manuscript we investigated the geographic aspect by looking at clustering of the studies based on the model coefficients (Fig. S8 in the original version, now Fig. S13 in the revised version). These earlier results however showed that geography alone (potentially confounded by technical differences that align with geography) does not explain the differences between the study-specific models. We have further investigated this aspect for the revision of our manuscript.

Action taken: Additional analyses incorporated in the revised version of our manuscript, first include a PERMANOVA analysis where we tested if geographic origin of the study (W vs E) affects microbiome structure. We added this information in Supplementary Table 1. This geographic grouping variable only explains a small fraction of variance when compared with the study of origin. Second, we divided the CSV values based on the E/W origin of the train and test dataset (see also comment above; Fig. S8). It can be seen that models built and evaluated on the E shotgun metagenomics cohorts generally perform a bit better, but this trend does neither generalise to W nor to the 16S datasets, suggesting that geography-associated effects are relatively small as in both cases the differences in AUCs are not statistically significant.

3) Although several microbiome signatures have been proposed to be commonly enriched or depleted in PD, the study did not propose a PD-specific ML models that helps to the differentiation of this disease from other neurodegenerative diseases.

Response: We agree with the reviewer that this is an important aspect. However, this is difficult to implement for a similar reason as outlined above relating to study heterogeneity. As discussed and shown also previously by our group, while one may be tempted to do this type of analysis by pooling data related to different diseases, training a model using only PD as cases, and all other conditions plus disease-free samples as controls, this is generally a bad idea, because the resulting models will greatly overfit study differences and not capture true disease differences well (see e.g. PMID 33785070, Fig. 2). Hence, to address this question properly and rigorously, datasets are needed in which multiple disease types were collected from the same population and processed with the same methods. Unfortunately, we are not aware of any such dataset of sufficient size to train and evaluate ML models on.

4) The abundance of lactic-acid producing bacteria (e.g. Lactobacillus sensu-lato, Bifidobacterium, and Ruthenibacterium) may not be attributed only to the use of L-dopa, more discussions are required to interpret these findings.

Response: We had reported in the original version of the manuscript several considerations on why the lactic acid bacteria might be increased in abundance in PD. As rightly pointed out by the reviewer we also had concluded that this cannot only be due to drug usage (see page 17 lines 21-31 in the original manuscript).

Action taken: *We have stressed even more that the increase in lactic acid bacteria cannot solely be explained by drug usage and that this needs further investigation in experimental model systems (page 12 lines 41-47 and page 13 lines 4-5).*

Reviewer #2 (Remarks to the Author):

Romano et al. present a comprehensive study on the application of machine learning (ML) techniques to diagnose Parkinson's disease (PD) using gut microbiome data. It explores the effectiveness of various ML models, comparing the diagnostic accuracy of taxonomic and functional profiles of the microbiome. The study also investigates the impact of data preprocessing methods, such as sequencing depth and normalization, on model performance. Additionally, it addresses the challenge of model generalizability across different datasets and the specificity of microbiome-based diagnostics in distinguishing PD from other neurodegenerative diseases. Finally, it identifies the gut microbial taxa and their functions that are strongly associated with PD. This review paper is interesting to me. I am quite open to looking at a revised version if the authors could address some major and minor issues in a satisfactory fashion, which I describe in more detail below.

Major issues:

1. Regarding the claim "The study of origin instead explains a considerably higher proportion of the data, 19.9% and 7.7% for the 16S and SMG data, respectively", are there any figures to visually demonstrate this? For example, the authors can generate a similar distance-based redundancy analysis while all data points are colored by the study of origin to show how the data from different study origins separate.

Response: *Thank you for this comment. We apologise for the lack of clarity regarding this aspect in our original submission. There, Figure 1 A and B already displayed all samples coloured by study and the proportion of variance explained by the disease status and the study of origin estimated via a PERMANOVA analysis. To make this clearer we have rearranged the figure in the revised version of the manuscript. In addition, we have performed additional PERMANOVA tests including geographical features and added this information in the Supplementary Material (Supplementary Data 1).*

Action taken: *We adjusted Figure 1 to improve clarity by making separate panels for the boxplots showing the distribution of samples along the first two ordination coordinates to improve visibility. We have done this for both the 16S and SMG data for both the unconstrained ordination showing some clustering by study and the study-constrained ordination showing (absence of) clustering by disease even when study heterogeneity is accounted for.*

2. The training of machine learning (ML) methods, such as random forests, is not clear to me. How do the authors select the training data versus the test data? Did they use the random data splits? Is the presented performance obtained by measuring the predictive power of the test data? Additionally, I don't know how the hyperparameters are determined exactly. Is the cross-validation of the training data used to select the hyperparameters? I think a better clarification of the training, validation, and test has to be included in the "Comparison of machine learning approaches" and the method sections.

Response: We agree that these are very important aspects of any machine-learning application to biological data and apologise that this was not clearer in our original submission.

Action taken: While we originally mostly referred to paper describing the SIAMCAT workflow which takes care of these issues (from our group PMID: 33785070), in the revised version of the paper, we amended the Methods section (page 17 lines 35-45, page 18 lines 4-11) to provide more of these crucial details about ML training, hyperparameter tuning and evaluation workflows. In particular, we clarified where default SIAMCAT workflows were used or where and how we tailored them specifically for this study.

3. I worry that the comparison between the average AUC (area under the receiver operating characteristics curve of predicting PD or control based on the 16S taxonomic profile and that based on the shotgun taxonomic profile in Fig. 2 might be heavily based on the difference in sample size across datasets. Presumably, a large dataset would have more samples to train the ML algorithms and thus lead to better predictive performance on the test data. Therefore, the comparison between two cases with different sample sizes can cause erroneous results. I would suggest they only select samples with paired 16S and shotgun metagenomic data and then use the matched samples with paired data to investigate if the predictive performance using the 16S taxonomic profile is different from that using the shotgun taxonomic profile.

Response: We agree with the reviewer that one expects training set size to have some impact on the resulting ML model accuracy. Therefore, already in the original submission we had included an analysis which illustrates that, contrary to this expectation, training set size in our study is insufficient to explain the differences among CV, CSV, and LOSO AUCs (pages 5 lines 30-32 and page 8 lines 2-4 in the original manuscript) – potentially due to the many other technical and biological differences between studies which affect model accuracy (some of which are discussed above).

Unfortunately, we cannot perform a matched comparison between 16S and shotgun metagenomics data sets as, to the best of our knowledge, only the study by Jo et al. included both 16S and metagenomics data. For this dataset, the metagenomic data yielded models with higher AUC even though there are fewer metagenomic than 16S samples. However, we agree that it is worthwhile to better report this phenomenon in the revised version of the manuscript.

Action taken: We have added an additional Supplementary display item in which we show that the AUCs are not strongly influenced by the training set size (Fig. S7).

4. Regarding the comparison of AUC for the within-study cross-validation (CV) with the AUC for the leave-one-study-out validation (LOSO), I wonder whether the test data for both cases are the same. Specifically, when the authors measure the CV predictive performance on the same dataset, in principle, they use a fraction of the dataset for the test. Is it the same as the test set used in the LOSO? They need to be the same so that the performance benchmarking is meaningful.

Response: *The within-study CV is performed by splitting the data within a specific study (in 10 folds; to minimise stochastic fluctuations, we repeated this CV ten times, newly resampling subsets each time), whereas the LOSO validation is instead performed by leaving the data of one study completely out to be used as holdout set for testing the model. Thus, indeed model evaluation is not completely identical. However, during 10-fold CV the whole dataset is used for testing (10% in each fold) so that a test prediction is made for each sample in that dataset (albeit not all by the same model; test predictions from the ten repeats of CV are averaged and used to calculate AUCs). This approach should render AUCs as comparable as possible between within-study CV and LOSO, as they are indeed both derived from the full data set.*

Action taken: *For clarity, we have amended the method section by describing in more detail the way the various approaches were carried out (page 17 lines 35-45, page 18 lines 4-11 and lines 35-44).*

5. It is great to see the comparison of AUC for CV with that for the LOSO. However, I think the usefulness of pooling data across studies needs to be investigated from a different perspective: is pooling more independent datasets across studies better than only pooling a few? I believe it is important to change the number of independent datasets combined as the training set and study how the AUC for CV changes as the number of independent datasets used in the pooling varies. In this way, it helps explain how many datasets across studies should be pooled to enhance the predictive performance on a separate dataset.

Response: *Thank you for this comment. We agree that this might be an interesting technical aspect to investigate further. Due to very high computational demands of this analysis, we have performed it on a subset of the datasets only.*

Action taken: *For the metagenomics datasets (not the 16S data though) we have performed 7 independent LOSO validations, each with different training sets. These training sets included a progressively increasing number of data sets (i.e. studies, increasing from 2 to 6), and we evaluated this approach for all possible (shotgun metagenomic) LOSO test sets, which resulted in a total of 57 different training sets. For each training set, we built a single Ridge regression model, which was then tested on the hold-out dataset. We then plotted the resulting AUCs against training set sizes (Fig. S12). Statistical testing suggests that an increase in training set size has a significant effect on model accuracy. However, the results displayed in this figure also reveal large variability in AUC depending on the test set and that in general increasing the number of studies combined during training seems to decrease the variability in accuracy between models. Our explanation for this would be that pooling a higher number of studies helps better capture microbiome heterogeneity across the PD spectrum corresponding to an increase of prediction accuracies with the number of studies pooled for training.*

Minor issues:

1. I am confused about why the p-values for all cases in Fig. 1 are 0.001. Should it be < 0.001? It would be better to show the raw p-values.
2. Some figure labels are wrongly referred to in the text. For example, in the section "Cross-study portability of the ML models," when the authors describe the results of 16S, they should refer to Fig. S3 instead of Fig. S4. Similarly, when they describe the results of

the shotgun taxonomic profile, they should refer to Fig. S4 instead of Fig. S5. Please check the figure labels throughout the article.

3. On page 8, “all Ridge models built on the 16 and SMG data” should be “all Ridge models built on the 16S and SMG data”.

4. It seems to me that Fig. 5 looks vertically compressed. Please find a way to fix this issue.

Response: *Thank you very much for your detailed comments; we apologise for the lack of clarity with respect to the mentioned points.*

Action taken: *We have adjusted the p-values from the PERMANOVA and addressed all textual issues identified by the reviewer. Finally we have improved the quality of all our display items, including Fig 5 as suggested.*

Reviewer #2 (Remarks on code availability):

The code itself looks fine. But it only contains the code used for data analysis. No data processing is included. Thus, it is hard for a scientist to leverage the code to deal with a similar scientific analysis from raw data. I would recommend including an example in the README file to elaborate how to perform a similar data analysis based on raw data.

Response: *We appreciate that this reviewer is making sure that code for reproducing our analysis is available. We made an effort to publish all code for the meta-analysis so that our approach might be re-used. However, the basic re-analysis of raw data is relying on more standardised tools and workflows, which are relatively simple to re-implement, but at the same time are often computationally demanding, so that running them requires HPC environments. For this reason we feel that there is limited value in sharing our code for that but we have included additional information as requested in the README of the github repository.*

Action taken: *We added information on the profiling workflows used for generating taxonomic and functional profiles to the README of the github repository.*

Reviewer #3 (Remarks to the Author):

In this article the authors, present an extensive meta-analysis of 16S and SMG data from numerous studies using a ML approach to explore the microbiome signature implicated in Parkinson’s disease. They provide several hypotheses while analyzing the signatures and overall models’s performance. The article is easily readable, although some of the figures could be improved.

- Figure 1, For better lisibility it may be useful to reduce size of the points and eventyally remove the outlines from the boxplots or lower their size. The first two dimensions of the MDS explain little and this could be emphasized in the text. Another comparative approach could be the intra-study vs inter-study sample differences... Have the authors evaluated the

impact of geography alone rather than the study? Have the authors explored the specific effect of preprocessing protocols, which is known to be a strong determinant of inter-study variability? They mention that they investigated batch effects in the 16S data. How were these batches identified?

Response: We thank the reviewer for this comment and agree that Fig. 1 could be improved. Similarly, we agree that it would be interesting to explore the effects of geography further using beta-diversity analysis. Some of these analyses had already been included in the original version of our manuscript (based on the coefficients obtained from the ML models Fig. S8 in the original version of the paper). In the revised version of the manuscript we have implemented additional tests to elucidate these differences further following the suggestions of the reviewer (see Supplementary Data 1 and Fig. S1).

If we understand the reviewer correctly that preprocessing protocol refers to the way 16S and metagenomic data was generated (including DNA extraction); we agree that this will have large effects on the gut microbiome and likely also on study-specific PD signatures. In our previous meta-analysis we investigated this in detail (PMID: 33692356). To minimise overlap, in the present manuscript we focused on reporting the overall effect of study heterogeneity in the context of ML applications. We feel that an in-depth investigation of how individual experimental procedures affect microbiome profiles and disease signatures is beyond the scope of this work.

Regarding the definition of batches in the 16S dataset, we defined each study as a batch. We clarified this in the Method section of the revised version.

Action taken: In response to this comment and those raised by Reviewer 1 (which are included above), we have modified Fig. 1 and have performed additional PERMANOVA tests for the country, continent, and Western vs Eastern origin of each study. These results are reported in Supplementary Data 1. In addition we have plotted Bray-Curtis dissimilarities between samples within studies and between studies as suggested by the reviewer (Fig. S1). Finally, we have clarified in the Methods section how batches were defined (page 19 line 32)

- Concerning Figure 2 the authors write that: "For both 16S and SMG data the LOSO model performances were significantly higher than those obtained through CSV". Are they referring to the gray points? This annotation can be confusing with the CV (blue) and could benefit from being added in the Figure's 2 legend. What is the CV schematic? 10-fold? If there is more than one fold, it would also be interesting to illustrate the variability using a violin plot as well? (This is done in Figure 4)

Response: We appreciate this constructive comment and have amended Figure 2 following the reviewer's suggestions. The CV schematic was ten times repeated 10-fold cross validation as we reported in the method section of the original manuscript. We clarified this aspect in the revised manuscript and added an additional display item following the reviewer's suggestion.

Action taken: We modified Fig. 2 to clarify the use of colours and additionally produced another plot, in which we report AUCs and standard deviations for the within-study CV and

the LOSO validation (Fig. S6). Moreover, we clarified our ML workflow in the Methods section of the manuscript (page 17 lines 35-45, page 18 lines 4-11 and lines 35-44).

- Figure 3 is difficult to read. Looking at the distribution of the violinplots I'm not sure that averaging results is the best way to present these results. There are clearly some models (datasets), which work much better than others. Are these 16S or SMG data/models? Until now the data results on data types were presented separately (after going through the text it looks like 16S) - this information could be specified also in the legend. The gray lines in the orange violinplots are barely visible. It could be interesting to add the AD and MS classification results in two other panels in Figure 2 and also the PD datasets in Figure 3 as a baseline. Moreover, these diseases and data could be introduced before in the manuscript.

Response: *We thank the reviewer for this comment. We have modified Fig. 3 (Fig. 4 in the revised version of the manuscript) following the reviewer's suggestions. The approach we present here is based on testing the ML models built for the PD datasets on data obtained from other neurological diseases. For this reason we did not train models to classify other neurological diseases, as this was outside the scope of our work. Here we only used ML models trained in 16S PD data sets, as all the data we obtained for other neurological data was also generated using 16S rRNA gene sequencing (publicly available shotgun data is still scarce). This was specified in the original version of the manuscript in page 9 lines 6-10 and page 19 lines 23-26. Our approach of assessing disease specificity of ML models is based on defining a 10% FPR cutoff in the respective PD models, hence for a disease-specific model, the FPR expectation on any other data set (that does not contain PD cases) would be 10%. We have now added a line in the figure to clarify this aspect and further explained our approach in the text.*

Action taken: *We have modified the display item (now Fig. 4) and its caption and clarified the approach we used in the Methods sections (page 19 lines 38-44).*

- The authors write: "In our meta-analysis, we found that models built on taxonomic profiles perform in general better than those built on functional profiles (Fig 4)". However from this figure the variability of the performance is large and it is not clear how statistically significant this is. Can the authors provide more statistics backing up this claim?

Response: *Thank you for the comment on this aspect of the manuscript. To incorporate this suggestion, we refined the way we compute ML models from KO profiles. We first performed a feature selection (based on univariate association with the label on the training set) and then built a ML model on this feature subset. We additionally tested for significant differences in AUC as a consequence of different feature types used, following the reviewer's suggestion.*

Action taken: *We further investigated the relationship between profile types across the different validation strategies using linear models. We then extracted all contrasts and included these results in Supplementary Data 3.*

- I understand that the feature spaces are different between WGS and 16S thus making comparison of the models difficult. But have the authors tried to bring these two spaces together?

Response: We appreciate the comment of the reviewer. Indeed taxonomic profiles differ substantially between 16S and WGS. Reconciling the respective feature spaces is however not straightforward but some advances have been made recently (PMID: 37500913). Our group is also investigating how to robustly integrate these two types of data. For this, we experimented with a comparable approach using the GTDB as a unifying taxonomy. However LOSO results based on unified (genus-level) taxonomic profiles (in which 16S and shotgun datasets were combined for training) generally looked worse than those that were based on more technically homogeneous (using either shotgun or 16S) datasets. We believe that this might be due to yet unresolved technical aspects in harmonising different types of taxonomies used for profiling 16S and SMG data. Hence, we believe that reporting these results will not accurately reflect the true value of merging data types. We therefore feel that substantially more effort is needed to make this work and that this is beyond the scope of this study.

- Figure S2, it is not clear whether the results illustrated here concern only for the two studies mentioned in the legend or all the studies? Idem for Figure S1.

Response: Thank you for this comment. One of these figures is based on all the 16S the other one on all SMG datasets included in our meta-analysis.

Action taken: We have clarified in the legend that these figures (in the revised Supplement Figures S2 and S3) refer to either all 16S amplicon datasets (Fig. S2) or all shotgun metagenomics dataset (Fig. S3).

- Figure S3 is difficult to see patterns, maybe some kind of hierarchical clustering could help improve visibility.

Response: We have tried to perform a clustering on the AUC values as suggested (see figure below). Unfortunately, the results do not seem to be very informative as they break the diagonal structure which is needed to understand the schematic of the cross-study validation. Hence, we believe that this type of display is not ideal.

We agree with the reviewer that it is of interest to understand how similar the models built on individual studies are. We would however argue that this question is better answered by ordination or clustering of the model coefficients as we had done in the original version of the manuscript (Fig. S8 in the original version and Fig S13 in the revised version of the manuscript). This ordination can also be examined to address the question of whether datasets collected on the same continent may give rise to more similar models compared to those built on data from different continents.

- Figure S6: please specify the metric used. What about the SMG dataset, what is the impact of rarefaction there? How does the rarefact fluctuate when changint the seed ?

Response: Thank you for this comment. We have amended the figure and its caption to specify the data we report on. We have not investigated rarefaction here or the impact different random seeds may have on downstream ML analyses, as we believe this to be a very technical exercise that likely yields limited insights compared to the enormous computational costs required to perform such an analysis: to address this, one might need 100s of rarefied data sets from which 1000s of models would need to be built. However, previous work has shown that rarefaction is a suboptimal preprocessing approach for statistical or ML analyses (e.g. PMIDs: 38054712, 38443997). Here, we have performed a small-scale analysis on the 16S datasets to verify these results (Fig S6 in the original version and Fig S10 in the revised version of the manuscript). Indeed, we did not observe a strong effect of rarefaction, and hence decided to not investigate this further.

- Figure S8: not sure that the model weights in the ridge algorithm is the best descriptor to compare datasets. A more consensual descriptor in terms of predictability could be the feature importance for instance. In this case the authors can not conclude much, since there are issues with the modeling itself. Also elipses could improve/facilitate the interpretation

Response: Thank you for this comment. We agree that many methods exist to evaluate feature importance for various ML models, which necessitates careful consideration. In particular for nonlinear models, many external estimation procedures have been developed. However, for the (linear) Ridge regression approach, we believe that the coefficients

themselves are a good representation of feature importance that can be extracted without additional computational costs. We also verified that Ridge model coefficients tend to agree well with SHAP values, which are a popular modern alternative for assessing feature importance. We additionally verified that the Ridge regression model coefficients indeed correlated with the results of our univariate analyses (showing that higher coefficients correspond to taxa significantly enriched in PD; Fig S19 in the revised version of the manuscript). In that sense we think that the Ridge coefficients are a good descriptor of the models built on the individual datasets, and can therefore be used to compare study-specific models. As we mostly intended to use Fig. S8 (Fig. S13 in the revised version of the manuscript) to compare model characteristics across datasets, we believe that the use of the Ridge coefficients is appropriate to capture this information.

Action taken: *We further validated that Ridge regression coefficients strongly correlate with the features detected to be associated with PD in the univariate analyses and have added a figure that illustrates this in the Supplementary materials (Fig. S19).*

- The co-variate analysis was restricted to age, sex, the authors also mentioned medication usage. Can they report for which percentage of the cohorts was this available and at what resolution ? The lack of data can explain the low rate of variables affected by these cofounding.

Response: *Thank you for this comment. In the original version of our article we have performed the co-variate analyses for sex, age, and PD and non-PD drug usage. We agree with the reviewer that the lack of metadata for all datasets might result in a limited number of features detected as potentially confounded. In the original version of the manuscript we have reported this in page 16 lines 22-38 and page 21 lines 44-47.*

Action taken: *We clarified the percentage of studies with available metadata in the method section in page 16 lines 40-41.*

- Figure 6b, please specify what the black rectangles indicate

Response: *Thank you for this comment and apologies for being unclear.*

Action taken: *We clarified this in the legend of Fig. 6 (Fig 7 in the revised version of the manuscript).*

- The interpretation of the results on the functional spaces, although interesting, is only based on data that indicate the functional potential, which is also difficult to estimate as the different modules and pathways may or may not be used to compute the relative abundance. It is important not to oversell here and prevent speculation.

Response: *We appreciate the reviewer's comment and agree that caution needs to be exercised when interpreting metagenomic data. While shotgun metagenomics data cannot directly lead to functional insights on the mechanisms underlying the interactions between microbiome and disease processes, they can help develop new hypotheses and guide further research. We have stressed this aspect throughout the manuscript and have pointed out in page 13 lines 17-18, page 14 line 1-2 and 39-42, page 17 lines 16-21, page 19 lines 5-*

9 of the original manuscript that integrative multi-omics approaches and additional experimental data are needed to really understand the functional role of the microbiome in PD. We carefully revised the text to avoid overselling our findings.

Action taken: We have revised the Discussion to make this clearer (pages 13 lines 4-5, lines 19-22, and 30-35, page 14 lines 19-24 of our revised manuscript).

- Can the authors comment on the soundness of the covariate analyses feature ~ sex + age + PD + (1|cohort), knowing the distribution of microbiome features ?

Response: Thank you for pointing this out. We agree that for microbiome data analysis it is not entirely clear how well parametric approaches like linear models work, given that the data (due to sparsity and compositionality) may not meet all their assumptions. For this reason, we have used a nonparametric approach to infer overall differential abundances between PD and controls in the paper. However, parametric approaches provide a great deal of flexibility to assess confounding effects of covariates, which would be otherwise difficult to implement. Moreover, there are now several studies (including one from our group) showing that linear models may be particularly suitable for confounder adjustment in microbiome data (PMIDs: 35134120, 34784344; DOI: 10.1101/2022.05.09.491139).

Action taken: To specifically verify that linear models are appropriate in the context of our meta-analysis, we selected a representative subset of models and looked at the overall distributions of the residuals. While it is clear that in some cases the residuals are not random or normally distributed, it is also evident that in many cases such deviations are tolerable or absent (see figure below). Overall, we conclude that in most cases the estimates should be reasonable.

- Concerning data availability, the authors state: "All data used in the article are either publicly available or have been directly obtained from the authors of the original publications, as specified in Table 1.". However, the authors have put considerable effort in analyzing

these public dataset and computing the different taxonomic and functional profiles. It would quite useful to the scientific community that these processed data (taxonomic, functional, metadata, and code) are easily accessible to the community, the same was as the original data were accessible to them.

Response: *Thank you for this comment. We indeed intend to make these data available.*

Action taken: *We prepared everything to make the data available after acceptance of the paper.*

Reviewer #3 (Remarks on code availability):

I briefly explore the code without testing it

REVIEWER COMMENTS

Reviewer #2 (Remarks to the Author):

I appreciate that the authors have responded satisfactorily to most of my comments. Yet, I still have concerns over their response to my previous major point 3. The presented Fig. S7 needs to be more convincing because different sample sizes across datasets cannot be used to support their argument that the training set size does not strongly influence the AUCs. Instead, I bet if they use the same dataset and vary the training sample size, the training sample size will strongly influence the AUC.

I understand that it is hard to have a matched comparison between 16S and shotgun metagenomics datasets. However, as the authors mentioned in their response, the Jo et al. dataset includes 16S and metagenomics data. I believe the authors can (1) find the shared samples with both 16S and metagenomics data, (2) leave the same samples out for the test data, and (3) then perform a one-to-one comparison between 16S and metagenomics data using the same training sample size. It is better to show this comparison directly as a figure to demonstrate their claim that using the metagenomic data can generate better predictive performance than using the 16S data.

Response: *Thank you very much for the comment. We agree with the reviewer that when a large homogeneous data set is downsampled to artificially reduce the training set size, ML accuracy will inevitably drop at some point. This is a well-known and often reported phenomenon. However, we tried to address a question that is more relevant to our meta-analysis setting, namely “In a heterogeneous setting, what is the relation between training set size and AUC and why do large data sets not always result in better AUCs?” by presenting the data shown in Supplementary Fig. 7. Here, we contrast study effects with training set size and show that the variation in AUC across study is so large that it overrides potential gains of accuracy that are theoretically expected for larger training sets. We believe that this is a more important aspect with significant practical implication in the microbiome field. Our findings here are in agreement with our earlier work on applying machine learning to microbiome data in large cross-study comparisons (Wirbel et al. *Genome Biol.* 2021. DOI: 10.1186/s13059-021-02306-1). Moreover, in Supplementary Fig. 7 it can be seen that the differences in sample sizes between shotgun metagenomics (SMG) and 16S are not drastic, and, if we exclude the largest dataset for both data types, 16S datasets are in general larger than those for SMG (average size 171 vs 100 samples). To further clarify this last point and make the performance of 16S and SMG-based models more comparable, we have performed a final additional experiment as suggested. In this analysis, we matched the 16S and the SMG data generated from the same samples in the study by Jo et al. and built new Ridge regression models by using exactly the same training and test splits in both 16S and SMG data.*

Action taken: *To avoid any misunderstanding about the effect of sample size on model accuracy in a heterogeneous meta-analysis setting and the theoretical expectation (for a homogeneous data set), we reworded the main text (page 5) and the caption of Supplementary Fig. 7. Moreover, we matched the samples between the 16S and the SMG data published by Jo et al and used exactly the same samples in the training and test splits to build additional Ridge regression models. Although we believe that the conclusions drawn from a single dataset are not as strong as those derived from a meta-analysis setting (shown in Supplementary Fig. 7), this additional analysis confirms the*

notion that the SMG-based models perform better than those based on 16S data (AUCs of 82.4% and 70.4% for SMG and 16S data, respectively). We have incorporated a statement describing this result in the revised version of the manuscript (pages 5 and 19, changes highlighted).